# Multi-Modal Interactive Agent Layer for Few-Shot Universal Cross-Domain Retrieval and Beyond

**Kaixiang Chen**[1,2]**, Pengfei Fang**[1,2*] **, Hui Xue**[1,2*]

[1]School of Computer Science and Engineering, Southeast University
[2]Key Laboratory of New Generation Artificial Intelligence Technology
and Its Interdisciplinary Applications (Southeast University), Ministry of Education, China
{kxchen, fangpengfei, hxue}@seu.edu.cn

## Abstract

This paper firstly addresses the challenge of few-shot universal cross-domain retrieval (FS-UCDR), enabling machines trained with limited data to generalize to novel retrieval scenarios, with queries from entirely unknown domains and categories. To achieve this, we first formally define the FS-UCDR task and propose the **M**ulti-Mod**a**l **I**nteractive Agent **L**ayer (MAIL), which enhances the cross-modal interaction in vision-language models (VLMs) by aligning the parameter updates of target layer pairs across modalities. Specifically, MAIL freezes the selected target layer pair and introduces a trainable agent layer pair to approximate localized parameter updates. A bridge function is then introduced to couple the agent layer pair, enabling gradient communication across modalities to facilitate update alignment. The proposed MAIL offers four key advantages: **1)** its cross-modal interaction mechanism improves knowledge acquisition from limited data, making it highly effective in low-data scenarios; **2)** during inference, MAIL integrates seamlessly into the VLM via reparameterization, preserving inference complexity; **3)** extensive experiments validate the superiority of MAIL, which achieves substantial performance gains over data-efficient UCDR methods while requiring significantly fewer training samples; **4)** beyond UCDR, MAIL also performs competitively on few-shot classification tasks, underscoring its strong generalization ability. **Code**.

## 1 Introduction

The objective of universal cross-domain retrieval (UCDR) [37, 3] is to retrieve images from the real world (*Real* domain) using queries originating from unseen domains and classes. To achieve robust performance in these generalized retrieval scenarios, UCDR methods typically require extensive, diverse, and well-annotated datasets from multiple domains to learn domain-agnostic feature embeddings [37, 43, 13]. However, labeling data across multiple domains in real-world scenarios is often prohibitively expensive. More critically, in the UCDR task, the substantial domain gap between training and testing domains implies that excessive reliance on source domain data may lead to overfitting and poor generalization to unseen domains. Given

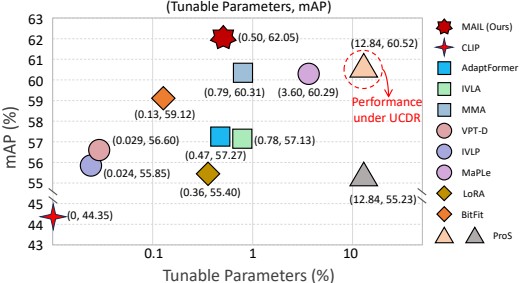

Figure 1: Comparison of MAIL with various methods on DomainNet [38] dataset under FS-UCDR (2-shot). The symbol □ indicates adapter-based methods, ○ represents prompt-based methods, ◇ denotes partially fine-tuned methods, and △ denotes ProS [13], the SOTA method for UCDR.

---

*Co-corresponding authors

39th Conference on Neural Information Processing Systems (NeurIPS 2025).

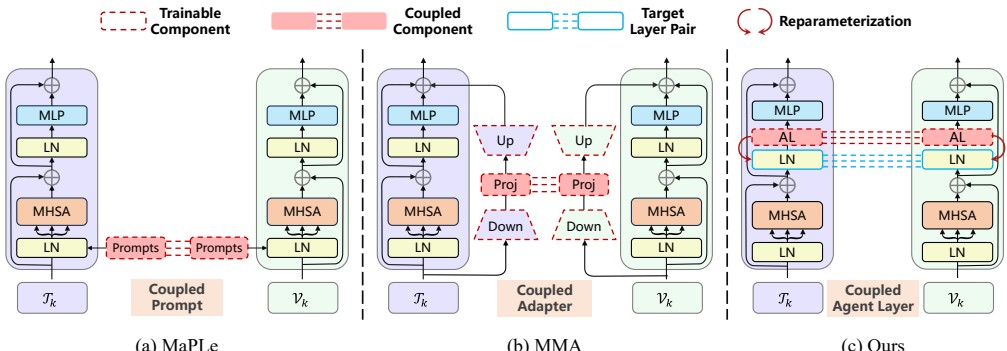

Figure 2: Modality-coupled methods in fine-tuning VLMs: (a) MaPLe achieves multi-modal alignment by establishing interconnections between the prompts. (b) MMA designs a unified feature-projection layer within the adapter that is shared by both modalities. (c) In contrast, MAIL achieves multi-modal alignment while preserving inference efficiency by introducing linked trainable agent layers (AL) that align parameter updates without altering the original model structure.

that recent CLIP-based methods [39, 24, 49] have demonstrated strong performance in few-shot classification, indicating the potential to generalize well with limited supervision, this raises a natural question: *Is it possible to train a model using only a few samples from each domain to achieve performance comparable to, or even surpass, existing data-efficient UCDR methods?*

In response, this paper formally defines the problem of few-shot UCDR (FS-UCDR), which aims to train a retrieval model using minimal samples per class from each source domain. To address this challenge, we explore parameter-efficient fine-tuning (PEFT) for pretrained vision-language models (VLMs) like CLIP [39], whose remarkable "zero-shot" generalization capabilities position them as promising tools for the FS-UCDR problem. We first empirically conduct a comprehensive empirical study on fine-tuning CLIP under the FS-UCDR setting, covering **four** categories of methods: **three** types of PEFT methods—❶ *adapter-based methods* [6, 49], ❷ *prompt-based methods* [23, 24], and ❸ *partially fine-tuned methods* (either through direct tuning such as BitFit [52] or indirect tuning such as LoRA [22])—as well as ❹ the previous state-of-the-art UCDR method, ProS [13]. The results are shown in **Fig. 1**, where we observe that modality-coupled methods (MCMs) (e.g., MaPLe [24] and MMA [49]; see **Fig. 2-a** and **Fig. 2-b**) consistently outperform their modality-independent counterparts, i.e., independent vision-language prompt / adapter (IVLP / IVLA). Since the objective of vision–language training is to achieve effective alignment between modalities, the explicit cross-modal interactions inherent in MCMs strengthen this alignment and facilitate knowledge transfer from source domains, which is particularly valuable in low-data scenarios. Furthermore, cross-modal interaction can be interpreted as a form of regularization: such coupling ensures that updates in one modality are propagated to the other, thereby promoting more coherent and consistent representations.

Building upon the three explored types of PEFT methods, the first two—adapter-based and prompt-based methods—both include modality-coupled variants. However, modality coupling in the third type of PEFT methods has never been explored. This is primarily due to the non-trivial nature of such an endeavor. For instance, establishing cross-modal bias interactions in BitFit [52] or introducing modality coupling into the low-rank matrices of LoRA [22] poses significant challenges in both design and implementation. Given the unique advantage of the third type of PEFT methods—introducing no additional inference cost—designing a MCM for this category remains an important and yet underexplored problem. **To fill this gap**, this paper propose the **M**ulti-Mod**a**l **I**nteractive Agent **L**ayer (MAIL), which, to the best of our knowledge, is the **first** CLIP-based modality-coupled method explicitly aimed at enhancing the alignment of updates of the internal parameters between image and text modalities within the backbone, as illustrated in **Fig. 2-c**.

Specifically, to facilitate alignment updates between a specific pair of layers, MAIL designates these as the **target layer** pair while keeping their original parameters frozen. For each target layer, we introduce a lightweight **agent layer** that approximates the localized parameter updates through training. Each agent layer consists of a scaling and a shifting component [31], which together capture the fine-grained adjustments of the corresponding target layer. To encourage cross-modal interaction, we further incorporate a bridge function that couples the agent layers, enabling gradient flow between modalities during tuning and thereby strengthening alignment. At inference, the agent

layers are seamlessly reparameterized into their corresponding frozen layers, ensuring that the overall complexity of CLIP remains unchanged.

Compared to MaPLe and MMA, MAIL achieves superior performance with higher parameter efficiency in FS-UCDR and even surpasses ProS using only $1/140$ of the training data, as shown in **Fig. 1**. To summarize, our main contributions are outlined as follows: ❶ We formally define the problem of FS-UCDR and explore the potential of leveraging the pretrained vision-language model (VLM), specifically CLIP, to effectively address this challenge. ❷ We experimentally find that the modality-coupled method works effectively under FS-UCDR due to the explicit cross-modal interaction. ❸ We introduce MAIL, a novel MCM crafted to optimize the alignment of partially parameter updates across image and text modalities. ❹ Extensive experiments on *three* FS-UCDR benchmarks and *eleven* few-shot classification datasets demonstrate that MAIL achieves state-of-the-art performance while maintaining superior parameter efficiency and CLIP's inference efficiency.

## 2  Related Work

**Universal Cross-Domain Retrieval.** Cross-domain retrieval (CDR) [26, 16] addresses the inherent limitations of uni-domain retrieval (UDR) [5, 4, 42, 47] by enabling retrieval across diverse domains. However, it typically assumes that the testing phase involves known domains and semantic classes. This assumption restricts its applicability in real-world scenarios, where models often encounter entirely new domains and classes. Therefore, universal cross-domain retrieval (UCDR) [37, 43] has emerged as a promising direction, which leverages queries from unseen domains and unseen classes to retrieve semantically similar examples from the *Real* domain. However, this setup demands substantial data collection from diverse domains, which is both costly and time-intensive. To address these challenges, we propose a more practical and significantly more challenging variant: few-shot universal cross-domain retrieval (FS-UCDR), where only a limited number of samples per class are available for training, aiming to reduce data requirements while preserve retrieval effectiveness.

**Parameter Efficient Fine-Tuning.** With the rapid advancement of datasets, model architectures, and training algorithms [51, 7], foundation models like BERT [10], ViT [11], and CLIP [39] have revolutionized deep learning. However, their increasing size presents challenges for fine-tuning due to high memory and computational demands. To address these issues, parameter-efficient fine-tuning (PEFT) methods have been proposed, which can be broadly categorized into three types: ❶*Prompt-based methods*. These methods introduce additional learnable tokens during fine-tuning, while keeping all other parameters fixed. These tokens can be integrated into the vision model [23], the language model [57], or both [24], depending on the specific task requirements. ❷ *Adapter-based methods*. These methods incorporates lightweight adapter modules that are updated during fine-tuning, leaving the original model parameters unchanged. Adapters can take various forms, such as bottleneck structures [21, 6], simple residual layers [15], or memory banks [55, 54], and can be implemented either sequentially [21] or in parallel [6] with the original model. ❸ *Partially fine-tuned methods*. Some of these methods *directly* update a limited (target) subset of pretrained parameters, such as specific layers [1, 28] or biases [52], thereby minimizing overhead. Additionally, methods such as LoRA [22] and VeRA [27] *indirectly* approximate the partial updates by introducing new trainable parameters, which are merged back via re-parameterization to preserve inference efficiency.

## 3  Preliminary

### 3.1  Problem Setting

**UCDR.** In universal cross domain retrieval (UCDR), the training setup includes $N_S \geq 2$ source domains, collectively represented as $\mathbb{D}_S = \{\mathcal{D}_S^j\}_{j=1}^{N_S}$. Each domain is defined as $\mathcal{D}_S^j = \{(x_i^j, y_i^j)\}_{i=1}^{P_j}$, where $x_i^j$ denotes the $i$-th image out of a total of $P_j$ images in the $j$-th source domain, and $y_i^j$ represents its class label from a **shared** label space $\mathcal{Y}_S$. Additionally, the *Real* domain $\mathcal{D}_R$, consists of real-word images, serves a dual purpose by contributing to both training and testing. Within $\mathcal{D}_R$, there are two distinct subdomains: $\mathcal{D}_R^+$ and $\mathcal{D}_R^-$. $\mathcal{D}_R^+$ is a subset belongs to $\mathbb{D}_S$, i.e., $\mathcal{D}_R^+ \in \mathbb{D}_S$. $\mathcal{D}_R^-$ serves as the **gallery set** during testing. In the test phase, a query set $\mathcal{D}_Q = \{(x_i^q, y_i^q)\}_{i=1}^{P_q}$ is also provided, containing $P_q$ samples drawn from an unseen domain and unseen classes. By default, the classes in $\mathcal{D}_R^-$ are identical to those in $\mathcal{D}_Q$, and we denote this scenario as $\mathrm{UnseenGallery}$. However, in a realistic scenario, $\mathcal{D}_R^-$ may include additional classes, such as those from the training phase, which is refereed to as $\mathrm{MixedGallery}$.

$\mathbf{U}^D\mathbf{CDR}$ and $\mathbf{U}^C\mathbf{CDR}$. Universal domain cross-domain retrieval ($\mathrm{U}^D\mathrm{CDR}$) and universal class cross-domain retrieval ($\mathrm{U}^C\mathrm{CDR}$) are two specialized variations of UCDR. In $\mathrm{U}^D\mathrm{CDR}$, the query class is encountered during training, while in $\mathrm{U}^C\mathrm{CDR}$, the query domain has been seen.

**Few-Shot Setup.** The aforementioned setups demand substantial data collection across diverse domains, which is costly and time-intensive. To this end, we propose a more practical few-shot setup. While the shared label space consists of $C$ classes, we limit it to only $k$ (a small number) shots per class for each domain, resulting in a total of $|\mathbb{D}_S| = N_S \times k \times C$ training samples.

## 3.2 Revisiting CLIP

CLIP [39] is a pretrained vision-language model (VLM) consisting of two encoders: a text encoder, denoted by $\mathcal{T}(\cdot)$, and an image encoder (ViT [11] as default), denoted by $\mathcal{V}(\cdot)$. Both encoders comprise $L$ transformer [45] blocks, represented as $\{\mathcal{T}_i\}_{i=1}^L$ and $\{\mathcal{V}_i\}_{i=1}^L$, respectively. For classification inference with $C$ classes, CLIP inserts all class names into a pre-defined text template, e.g., "`a photo of a <category>`", generating $C$ inputs $\{t_i\}_{i=1}^C$ for the text encoder $\mathcal{T}(\cdot)$. For a certain input $t_y$, its output $\mathcal{T}(t_y)$ is:

$$\begin{aligned}
\mathcal{W}_0 &= \text{TextEmbed}(t_y) \\
\mathcal{W}_i &= \mathcal{T}_i(\mathcal{W}_{i-1}), \quad i = 1, 2, ..., L \\
\mathcal{T}(t_y) &= \text{TextProj} \circ \text{LN}(w_L^{N_t}),
\end{aligned} \tag{1}$$

where $\mathcal{W}_0 = [w_0^0, w_0^1, ..., w_0^{N_t}]^\top \in \mathbb{R}^{N_t \times d_t}$ is the word embedding, with $N_t$ and $d_t$ indicate text embedding length and dimension, $\circ$ represents the composition of functions. TextProj is a linear layer ($d_t \to d_t$). Similarly, for the image $I$, its representation $\mathcal{V}(I)$ is calculated as:

$$\begin{aligned}
\mathcal{P}_0 &= \text{PatchEmbed}(I) \\
[c_i, \mathcal{P}_i] &= \mathcal{V}_i([c_{i-1}, \mathcal{P}_{i-1}]), \quad i = 1, 2, ..., L \\
\mathcal{V}(I) &= \text{ImageProj} \circ \text{LN}(c_L),
\end{aligned} \tag{2}$$

where $\mathcal{P}_0 \in \mathbb{R}^{N_v \times d_v}$ is the image embedding, with $N_v$ and $d_v$ indicate embedding length and dimension, and $c_0 \in \mathbb{R}^{d_v}$ is the initial CLS Token. ImageProj is also a linear layer ($d_v \to d_t$). With $\mathcal{V}(I)$ available, the text features of the text templates with class labels are matched using the formula $p(y|I) = \frac{\exp(sim(\mathcal{T}(t_y), \mathcal{V}(I)))}{\sum_{i=1}^C \exp(sim(\mathcal{T}(t_i), \mathcal{V}(I)))}$, where $y \in \{1, 2, ..., C\}$, and $sim(.,.)$ refers to cosine similarity.

# 4 Multi-Modal Interactive Agent Layer

Modality-coupled methods (MCMs), such as MaPLe [24] and MMA [49], improve cross-modal alignment by enhancing synergy between encoders during training, thereby boosting retrieval performance under FS-UCDR. However, these benefits come with increased inference complexity. In contrast, the third type of PEFT methods—partially fine-tuned approaches—offers a key advantage: they introduce no additional inference overhead. Despite this, modality coupling has not been explored in this category. To bridge this gap, we propose the **M**ulti-Mod**a**l **I**nteractive Agent **L**ayer (MAIL), a lightweight MCM that aligns parameter updates across modalities while preserving the inference efficiency. As illustrated in **Fig. 3**, MAIL incorporates agent layers to capture localized parameter updates, while the bridge functions further refine and align these updates across encoders.

## 4.1 Agent Layer

The agent layer (AL) is designed to approximately capture the updates of specific operations within the encoders. It consists of **a scaling vector** $a$, initialized as an **all-one** vector, and **a shifting vector** $b$, initialized as an **all-zero** vector [31]. The agent layer can be appended after various positions:

$$\text{AL} \circ \text{OP}(x) = \text{OP}(x) \cdot \Lambda(a) + b, \tag{3}$$

where OP is a specific operation or a layer, $\Lambda(a)$ represents the diagonal matrix with the vector $a$ as its diagonal elements, $\cdot$ denotes the matrix multiplication. In a transformer block, the agent layer can be positioned after the LayerNorm (LN) layer (***Position-1***) to capture updates related to the parameters of LN. Similarly, it can be placed after the multi-head self-attention (MHSA) layer (***Position-2***) to monitor updates to the output weight matrix $W^O$, or after the MLP layer (***Position-3***) to track changes in the second linear layer, $W_{mlp}^2$. Beyond the confines of transformer blocks, the agent layer can also be appended after the final LN layer (***Position-4***) and the last projection layer

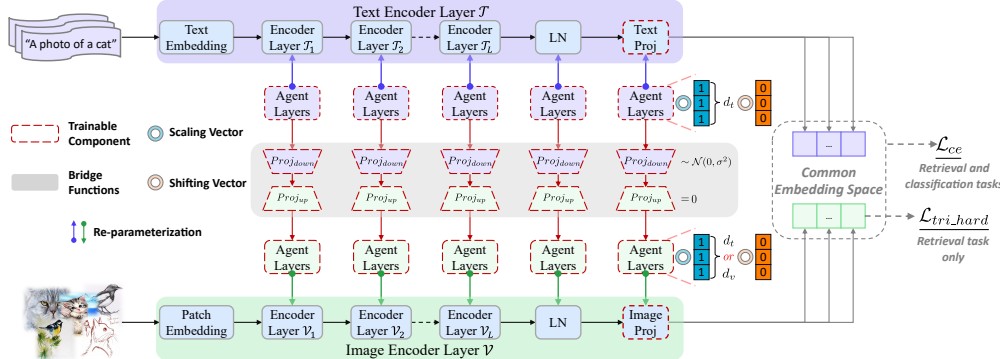

Figure 3: The proposed Multi-Modal Interactive Agent Layer (MAIL) for the transformer-based CLIP models. During training, we only fine-tune the agent layers, which are inserted into both encoders. The image agent layers interact with the text agent layers through a trainable bottleneck-based bridge function, fostering mutual synergy between the two modalities.

$W_{proj}$ (***Position-5***), effectively capturing the relevant updates. One can refer to **Appendix G.4** for a clearer visual illustration. In summary, the agent layer is specifically designed to track updates for **five positions** from two types of layers: the LN layer and the linear layer. For illustrative purposes, we will focus on the text encoder (i.e., $a, b \in \mathbb{R}^{d_t}$) to demonstrate how updates occur in the LN and linear layers.

**LN Layer:** The LN operation is formulated as:

$$\text{LN}(x) = \frac{x - \mu}{\sigma} \odot \gamma + \beta, \tag{4}$$

where $x \in \mathbb{R}^{N_t \times d_t}$ denotes the input to the LN layer, $\mu, \sigma \in \mathbb{R}^{N_t}$, $\gamma, \beta \in \mathbb{R}^{d_t}$. The agent layer, appended after the LN layer, can be formulated as:

$$\text{AL} \circ \text{LN}(x) = \frac{x - \mu}{\sigma} \odot \underbrace{\gamma \odot \underline{a}}_{\gamma} + \underbrace{\beta \odot \underline{a} + \underline{b}}_{\beta}, \tag{5}$$

where underline indicates the trainable component, and $\odot$ represents the Hadamard product. As shown in **Eq.** (5), the update of the agent layer can approximately correspond to the updates of $\gamma$ and $\beta$ in the LN layer.

**Linear Layer:** The linear layer is formulated as:

$$\text{LiL}(x) = x \cdot W^\top + bias, \tag{6}$$

where $x \in \mathbb{R}^{N_t \times d_a}$ represents the input, with $d_a$ can either equal to $d_t$ or the intermediate dimension of the MLP layer within the transformer block. $W \in \mathbb{R}^{d_t \times d_a}$ denotes the weight matrix, and $bias \in \mathbb{R}^{d_t}$ represents the bias. The agent layer, appended after the linear layer, is expressed as:

$$\text{AL} \circ \text{LiL}(x) = x \cdot \underbrace{(\Lambda(\underline{a}) \cdot W)^\top}_{W} + \underbrace{bias \odot \underline{a} + \underline{b}}_{bias}. \tag{7}$$

Due to the left multiplication by $\Lambda(a)$ applied to $W$, the updates of $W$ are row-wise.

During inference, based on **Eq.** (5) and **Eq.** (7), the agent layers can be seamlessly integrated into the original foundation model, **eliminating additional inference latency.**

### 4.2 Agent Layer Coupling

We define the text agent layer as $\text{AL}_{\text{text}} = \{a_t, b_t \in \mathbb{R}^{d_t}\}$, and the image agent layer at the same location in the image encoder as $\text{AL}_{\text{image}} = \{a_v, b_v \in \mathbb{R}^{d_v}\}$ or $\text{AL}_{\text{image}} = \{a_v, b_v \in \mathbb{R}^{d_t}\}$ (when the agent layer is appended after the final projection layer). These agent layers can be trained independently, and we term such a design as *independent vision-language updating* (IVLU). To

Table 1: FS-UCDR (2-shot) evaluation results (%) on DomainNet. * denotes the results are obtained using the full training data, i.e., the UCDR results. The best performance under FS-UCDR is marked as **bold** and the second best performance is marked as underline, while scores from our method are highlighted with a light purple background.

| Methods | Sketch | | | | Quickdraw | | | | Painting | | | |
|---|---|---|---|---|---|---|---|---|---|---|---|---|
| | UnseenGallery | | MixedGallery | | UnseenGallery | | MixedGallery | | UnseenGallery | | MixedGallery | |
| | $mAP_{200}$ | $Prec_{200}$ | $mAP_{200}$ | $Prec_{200}$ | $mAP_{200}$ | $Prec_{200}$ | $mAP_{200}$ | $Prec_{200}$ | $mAP_{200}$ | $Prec_{200}$ | $mAP_{200}$ | $Prec_{200}$ |
| ProS* [CVPR'24] | 64.67 | 60.01 | 58.43 | 54.63 | 28.42 | 25.44 | 23.18 | 21.27 | 75.16 | 69.55 | 71.20 | 66.12 |
| CLIP [ICML'21] | 42.20 | 35.28 | 36.62 | 29.79 | 7.44 | 5.61 | 6.00 | 3.17 | 61.68 | 55.07 | 56.53 | 50.14 |
| BitFit [ACL'22] | 62.68 | 57.71 | 55.84 | 51.52 | 24.88 | 21.99 | 19.22 | 16.93 | 73.40 | 67.39 | 68.46 | 62.93 |
| LoRA [ICLR'22] | 54.85 | 49.23 | 48.71 | 43.33 | 22.16 | 18.10 | 17.73 | 14.21 | 71.46 | 65.10 | 66.81 | 60.81 |
| VPT-D [ECCV'22] | 57.47 | 52.58 | 50.34 | 45.92 | 24.26 | 21.68 | 18.45 | 16.77 | 72.05 | 65.88 | 67.25 | 61.42 |
| AFormer [NeurIPS'22] | 59.64 | 53.73 | 52.91 | 47.77 | 23.11 | 19.86 | 18.11 | 15.71 | 71.08 | 64.59 | 66.14 | 60.17 |
| IVLP [CVPR'23] | 56.25 | 51.48 | 49.37 | 45.06 | 21.07 | 18.97 | 15.67 | 14.13 | 71.47 | 65.51 | 66.40 | 60.86 |
| IVLA [CVPR'24] | 60.01 | 54.16 | 53.27 | 48.19 | 23.96 | 20.72 | 18.74 | 16.38 | 71.02 | 64.60 | 66.10 | 60.18 |
| MaPLe [CVPR'23] | 62.50 | 57.49 | 55.69 | 51.67 | 27.57 | 24.93 | 21.58 | 19.75 | 75.02 | 69.41 | 70.61 | 65.38 |
| MMA [CVPR'24] | 63.86 | 58.84 | 56.74 | 52.33 | 27.37 | 24.28 | 21.84 | 19.56 | 74.08 | 68.36 | 69.24 | 63.92 |
| **MAIL** [Ours] | **65.76** | **61.57** | **59.05** | **55.25** | **29.41** | **26.95** | **22.83** | **21.26** | **76.05** | **70.85** | **71.12** | **66.44** |

| Methods | Infograph | | | | Clipart | | | | Average | | | |
|---|---|---|---|---|---|---|---|---|---|---|---|---|
| | UnseenGallery | | MixedGallery | | UnseenGallery | | MixedGallery | | UnseenGallery | | MixedGallery | |
| | $mAP_{200}$ | $Prec_{200}$ | $mAP_{200}$ | $Prec_{200}$ | $mAP_{200}$ | $Prec_{200}$ | $mAP_{200}$ | $Prec_{200}$ | $mAP_{200}$ | $Prec_{200}$ | $mAP_{200}$ | $Prec_{200}$ |
| ProS* [CVPR'24] | 57.98 | 54.42 | 52.19 | 49.56 | 76.48 | 71.86 | 72.28 | 68.15 | 60.52 | 56.26 | 55.46 | 51.95 |
| CLIP [ICML'21] | 50.08 | 44.74 | 43.75 | 38.91 | 60.37 | 51.30 | 56.08 | 46.91 | 44.35 | 38.40 | 39.80 | 33.78 |
| BitFit [ACL'22] | 58.84 | 55.14 | 52.32 | 48.86 | 75.81 | 70.27 | 71.04 | 65.85 | 59.12 | 54.50 | 53.38 | 49.22 |
| LoRA [ICLR'22] | 58.01 | 53.84 | 51.86 | 48.20 | 70.52 | 64.11 | 66.00 | 59.63 | 55.40 | 50.08 | 50.22 | 45.24 |
| VPT-D [ECCV'22] | 56.28 | 52.18 | 49.61 | 45.92 | 72.96 | 67.82 | 68.02 | 63.04 | 56.60 | 52.20 | 50.73 | 46.61 |
| AFormer [NeurIPS'22] | 59.55 | 55.00 | 53.17 | 49.27 | 73.19 | 66.28 | 68.66 | 62.1 | 57.27 | 51.89 | 51.79 | 47.00 |
| IVLP [CVPR'23] | 58.20 | 54.07 | 51.60 | 47.89 | 72.28 | 66.98 | 67.64 | 62.47 | 55.85 | 51.42 | 50.13 | 46.08 |
| IVLA [CVPR'24] | 57.39 | 53.36 | 50.86 | 47.22 | 73.31 | 66.40 | 68.75 | 62.21 | 57.13 | 51.84 | 51.54 | 46.68 |
| MaPLe [CVPR'23] | 59.45 | 56.14 | 53.05 | 49.95 | 76.92 | 72.28 | 71.98 | 67.62 | 60.29 | 56.05 | 54.58 | 50.87 |
| MMA [CVPR'24] | 59.33 | 55.01 | 53.11 | 49.23 | 76.95 | 71.62 | 72.10 | 67.09 | 60.31 | 55.62 | 54.60 | 50.42 |
| **MAIL** [Ours] | **60.11** | **57.40** | **53.34** | **50.95** | **78.94** | **74.80** | **73.91** | **70.14** | **62.05** | **58.31** | **56.05** | **52.80** |

enhance the synergy between the vision and language agent layers, we introduce a multi-modal agent layer coupling approach. Specifically, the scaling vector $a_v$ in the vision agent layer is integrated with $a_t$ via a bottleneck-based language-to-vision projection, acting as a bridge function that facilitates gradient exchange and promotes aligned updates across the modalities:

$$\bar{a}_v = a_v + W_{up} \cdot W_{down} \cdot a_t, \tag{8}$$

where $W_{up} \in \mathbb{R}^{d_v \times r}$ or $W_{up} \in \mathbb{R}^{d_t \times r}$ and $W_{down} \in \mathbb{R}^{r \times d_t}$, with $r$ representing the rank of the bridge function. Following the initialization method in LoRA [22], $W_{down}$ is initialized with random Gaussian values, i.e., $W_{down} \sim \mathcal{N}(0, \sigma_t^2)$, with $\sigma_t = \frac{1}{\sqrt{d_t}}$, while $W_{up}$ is initialized to zeros. $\bar{a}_v$ will replace in $a_v$ in the image agent layer. **Alg. 1, 2, 3** provides the pseudo-codes for MAIL in a PyTorch-like style. With just a few lines, MAIL can significantly boost performance in a plug-and-play manner. Additional design details and variants are provided in **Appendix G**, including the structure of the bridge function, various initialization strategies, pseudocode, and other implementation choices.

### 4.3 Parameter Analysis

Here, we analyze the parameter complexity of MaPLe, MMA, and the proposed MAIL. For a transformer block, MAIL fine-tunes $(8 + 4r) \cdot (d_t + d_v)$ parameters, where $r$ is set to 8 in our implementation for FS-UCDR. In contrast, MMA fine-tunes $2r_1 \cdot (d_t + d_v) + r_1^2$, with $r_1 = 32$ as the intermediate dimension of the bottleneck layer. Meanwhile, MaPLe requires $d_t d_v + N_p d_t$ parameters, with $N_p$ denotes the number of text prompts. Based on these calculations, we conclude that MAIL and MMA have a comparable number of learnable parameters, while both are significantly more parameter-efficient than MaPLe, i.e., MAIL $\sim$ MMA $\ll$ MaPLe. The detailed time and resource consumption are provided in **Appendix E**.

## 5 Experiments

In this section, we evaluate the effectiveness of our proposed MAIL on two tasks: **FS-UCDR** (including its variants) and **few-shot classification**. Details on **datasets, metrics, implementations, loss functions and computational cost** are provided in **Appendix A, B, C, D, E**.

Table 2: FS-U$^D$CDR (2-shot) evaluation results (%) on DomainNet.

| Methods | Sketch | | Quickdraw | | Painting | | Infograph | | Clipart | | Average | |
|---|---|---|---|---|---|---|---|---|---|---|---|---|
| | mAP$_{200}$ | Prec$_{200}$ | mAP$_{200}$ | Prec$_{200}$ | mAP$_{200}$ | Prec$_{200}$ | mAP$_{200}$ | Prec$_{200}$ | mAP$_{200}$ | Prec$_{200}$ | mAP$_{200}$ | Prec$_{200}$ |
| ProS* [CVPR'24] | 73.85 | 49.11 | 28.89 | 11.86 | 72.27 | 46.15 | 60.56 | 39.62 | 81.05 | 52.98 | 63.32 | 39.94 |
| CLIP [ICML'21] | 47.60 | 28.71 | 8.67 | 4.50 | 55.69 | 31.70 | 47.56 | 29.36 | 55.81 | 31.10 | 43.07 | 25.07 |
| BitFit [ACL'22] | 70.20 | 45.00 | 22.75 | 9.52 | 68.54 | 41.66 | 59.77 | 38.51 | 75.17 | 47.09 | 59.29 | 36.36 |
| LoRA [ICLR'22] | 58.01 | 62.05 | 39.23 | 17.67 | 7.34 | 65.72 | 38.90 | 57.94 | 35.52 | 41.77 | 54.59 | 32.55 |
| VPT-D [ECCV'22] | 65.63 | 42.96 | 22.33 | 10.01 | 66.66 | 41.13 | 57.63 | 37.20 | 73.64 | 46.46 | 57.18 | 35.55 |
| AFormer [NeurIPS'22] | 66.28 | 39.91 | 20.66 | 7.88 | 64.77 | 37.21 | 59.55 | 36.70 | 70.51 | 52.16 | 56.35 | 34.77 |
| IVLP [CVPR'23] | 64.54 | 42.17 | 21.31 | 9.51 | 66.55 | 40.58 | 59.26 | 37.64 | 72.31 | 44.80 | 56.79 | 34.94 |
| IVLA [CVPR'24] | 66.58 | 40.18 | 21.44 | 8.20 | 64.84 | 37.31 | 58.42 | 37.58 | 70.66 | 42.29 | 56.39 | 33.11 |
| MaPLe [CVPR'23] | 71.18 | 46.65 | 26.48 | 11.28 | 70.72 | 44.64 | 60.24 | 39.40 | 77.47 | 49.15 | 61.22 | 38.22 |
| MMA [CVPR'24] | 71.14 | 45.54 | 24.03 | 10.17 | 68.66 | 41.53 | 59.52 | 36.74 | 75.92 | 46.38 | 59.86 | 36.07 |
| **MAIL** [Ours] | **73.61** | **49.20** | **26.91** | **11.60** | **72.53** | **45.93** | **62.69** | **41.35** | **79.81** | **52.21** | **63.11** | **40.06** |

## 5.1 Experimental Setup for FS-UCDR and Its Variants

In terms of the retrieval task, we conduct three core evaluations to comprehensively assess MAIL's performance: FS-UCDR, FS-U$^D$CDR, and FS-U$^C$CDR. All experiments utilize a 2-shot setting, i.e., only $2 \times N_S$ training examples per category, where $N_S$ denotes the number of source domains.

**Datasets.** We conduct experiments on three benchmark datasets: DomainNet [38], Sketchy [41, 32], and TU-Berlin [12, 53]. DomainNet is utilized for FS-UCDR and FS-U$^D$CDR evaluations, while the Sketchy and TU-Berlin datasets are employed for FS-U$^C$CDR evaluation.

**Baselines.** We primarily compare our methods with the following categories: ❶ adapter-based methods, including AdaptFormer (vision-only adapter) [6], IVLA (modality-independent adapter) [49], and MMA [49]; ❷ prompt-based methods, including VPT-D (vision-only prompt) [23], IVLP (modality-independent prompt) [24], and MaPLe [24]; ❸ partially fine-tuned methods, including LoRA [22] and BitFit [52]; ❹ other methods, including the zero-shot CLIP [39] and the SOTA method under UCDR, ProS [13]. Implementation details of these methods are provided in **Appendix F**.

Table 3: FS-U$^C$CDR (2-shot) evaluation results (%) on Sketchy and TU-Berlin.

| Methods | Sketchy | | TU-Berlin | |
|---|---|---|---|---|
| | mAP$_{200}$ | Prec$_{200}$ | mAP$_{all}$ | Prec$_{100}$ |
| ProS* [CVPR'24] | 69.91 | 65.45 | 66.75 | 74.42 |
| CLIP [ICML'21] | 35.82 | 33.08 | 31.45 | 46.12 |
| BitFit [ACL'22] | 67.71 | 64.01 | 65.51 | 73.68 |
| LoRA [ICLR'22] | 54.23 | 52.77 | 57.78 | 67.97 |
| VPT-D [ECCV'22] | 65.19 | 61.16 | 62.12 | 70.89 |
| AFormer [NeurIPS'21] | 56.87 | 52.31 | 58.95 | 70.14 |
| IVLP [CVPR'23] | 60.27 | 55.75 | 59.13 | 68.58 |
| IVLA [CVPR'24] | 56.77 | 52.71 | 59.13 | 70.42 |
| MaPLe [CVPR'23] | 71.86 | 68.14 | 65.90 | 73.73 |
| MMA [CVPR'24] | 61.59 | 57.14 | 63.70 | 72.69 |
| **MAIL** [Ours] | **73.46** | **69.73** | **67.97** | **75.10** |

## 5.2 Experimental Setup for Few-Shot Classification

We also conduct three core few-shot classification evaluations that are widely adopted in prior work: ❶ base-to-novel generalization, ❷ cross-dataset evaluation, and ❸ domain generalization. All experiments are conducted under a 16-shot setting, i.e., using 16 training examples per category.

**Datasets.** We conduct the base-to-novel generalization and cross-dataset evaluations across 11 diverse image classification datasets: ImageNet [9], Caltech101 [14], OxfordPets [36], StanfordCars [29], Flowers102 [34], Food101 [2], FGVCAircraft [33], SUN397 [48], UCF101 [35], DTD [8], and EuroSAT [17]. In terms of the domain generalization evaluation, we use ImageNet as the training dataset and evaluate on four variants—ImageNetV2 [40], ImageNet-Sketch [46], ImageNet-A [19], and ImageNet-R [18]—each introducing different types of domain variation.

**Baselines.** We primarily compare our method with prompt-based and adapter-based approaches, including CoOp [57], CoCoOp [56], MaPLe [24], and MMA [49], as well as regularization-based methods such as KgCoOp [50], PromptSRC [25], and DeKg [30]. To ensure fairness, we exclude methods that rely on LLMs or adopt regularization purely as an auxiliary loss trick.

## 5.3 Comparison Results

**Comparison Results under FS-UCDR and FS-U$^D$CDR.** We compare the FS-UCDR and FS-U$^D$CDR performance of our MAIL against other baselines on DomainNet, as summarized in **Tab. 1** and **Tab. 2**. We identify several key observations: ❶ *Our method consistently outperforms existing baselines.* Notably, under FS-UCDR, MAIL significantly outperforms ProS, a model operate under data-efficient UCDR. Moreover, MAIL achieves comparable performance under FS-U$^D$CDR while

Table 4: Base-to-novel generalization (16-shot) evaluation results (%) across 11 datasets.

| Methods | Average | | | ImageNet | | | Caltech101 | | | OxfordPets | | |
|---|---|---|---|---|---|---|---|---|---|---|---|---|
| | Base | Novel | HM | Base | Novel | HM | Base | Novel | HM | Base | Novel | HM |
| CLIP [ICML'21] | 69.34 | 74.22 | 71.70 | 72.43 | 68.14 | 70.22 | 96.84 | 94.00 | 95.40 | 91.17 | 97.26 | 94.12 |
| CoOp [IJCV'22] | 82.69 | 63.22 | 71.66 | 76.47 | 67.88 | 71.92 | 98.00 | 89.81 | 93.73 | 93.67 | 95.29 | 94.47 |
| CoOpOp [CVPR'22] | 80.47 | 71.69 | 75.83 | 75.98 | 70.43 | 73.10 | 97.96 | 93.81 | 95.84 | 95.20 | 97.69 | 96.43 |
| KgCoOp [CVPR'23] | 80.73 | 73.60 | 77.00 | 75.83 | 69.96 | 72.78 | 97.72 | 94.39 | 96.03 | 94.65 | 97.76 | 96.18 |
| MaPLe [CVPR'21] | 82.28 | 75.14 | 78.55 | 76.66 | 70.54 | 73.47 | 97.74 | 94.36 | 96.02 | 95.43 | 97.76 | 96.58 |
| PromptSRC [ICCV'23] | 84.26 | 76.10 | 79.97 | 77.60 | 70.73 | 74.01 | 98.10 | 94.03 | 96.02 | 95.33 | 97.30 | 96.30 |
| MMA [CVPR'24] | 83.20 | 76.80 | 79.87 | 77.31 | 71.00 | 74.02 | 98.40 | 94.00 | 96.15 | 95.40 | 98.07 | 96.72 |
| DeKg [ICLR'25] | 84.96 | 76.38 | 80.44 | 77.40 | 69.20 | 73.07 | 98.64 | 95.20 | 96.89 | 94.47 | 97.76 | 96.09 |
| **MAIL** [Ours] | **85.19** | **77.39** | **81.10** | **77.92** | **71.22** | **74.42** | 98.34 | **95.36** | 96.83 | **95.50** | 97.97 | **96.72** |

| Methods | StanfordCars | | | Flowers102 | | | Food101 | | | FGVCAircraft | | |
|---|---|---|---|---|---|---|---|---|---|---|---|---|
| | Base | Novel | HM | Base | Novel | HM | Base | Novel | HM | Base | Novel | HM |
| CLIP [ICML'21] | 63.37 | 74.89 | 68.65 | 72.08 | 77.80 | 74.83 | 90.10 | 91.22 | 90.66 | 27.19 | 36.29 | 31.09 |
| CoOp [IJCV'22] | 78.12 | 60.40 | 68.13 | 97.60 | 59.67 | 74.06 | 88.33 | 82.26 | 85.19 | 40.44 | 22.30 | 28.75 |
| CoOpOp [CVPR'22] | 70.49 | 73.59 | 72.01 | 94.87 | 71.75 | 81.71 | 90.70 | 91.29 | 90.99 | 33.41 | 23.71 | 27.74 |
| KgCoOp [CVPR'22] | 71.76 | 75.04 | 73.36 | 95.00 | 74.73 | 83.65 | 90.50 | 91.70 | 91.09 | 36.21 | 33.55 | 34.83 |
| MaPLe [CVPR'23] | 72.94 | 74.00 | 73.47 | 95.92 | 72.46 | 82.56 | 90.71 | 92.05 | 91.38 | 37.44 | 35.61 | 36.50 |
| PromptSRC [ICCV'23] | 78.27 | 74.97 | 76.58 | 98.07 | 76.50 | 85.95 | 90.67 | 91.53 | 91.10 | 42.73 | 37.87 | 40.15 |
| MMA [CVPR'24] | 78.50 | 73.10 | 75.70 | 97.77 | 75.93 | 85.48 | 90.13 | 91.30 | 90.71 | 40.57 | 36.33 | 38.33 |
| DeKg [ICLR'25] | 81.18 | 74.75 | **77.83** | 98.58 | 75.18 | 85.30 | 90.73 | 91.55 | 91.14 | 45.20 | 35.09 | 39.51 |
| **MAIL** [Ours] | **82.27** | 72.03 | 76.81 | 98.20 | 75.27 | 85.22 | 90.54 | 91.77 | 91.15 | **47.80** | 36.27 | **41.24** |

| Method | SUN397 | | | DTD | | | EuroSAT | | | UCF101 | | |
|---|---|---|---|---|---|---|---|---|---|---|---|---|
| | Base | Novel | HM | Base | Novel | HM | Base | Novel | HM | Base | Novel | HM |
| CLIP [ICML'21] | 69.36 | 75.35 | 72.23 | 53.24 | 59.90 | 56.37 | 56.48 | 64.05 | 60.03 | 70.53 | 77.50 | 73.85 |
| CoOp [IJCV'22] | 80.60 | 65.89 | 72.51 | 79.44 | 41.18 | 54.24 | 92.19 | 54.74 | 68.69 | 84.69 | 56.05 | 67.46 |
| CoOpOp [CVPR'22] | 79.74 | 76.86 | 78.27 | 77.01 | 56.00 | 64.85 | 87.49 | 60.04 | 71.21 | 82.33 | 73.45 | 77.64 |
| KgCoOp [CVPR'22] | 80.29 | 76.53 | 78.36 | 77.55 | 54.99 | 64.35 | 85.64 | 64.34 | 73.48 | 82.89 | 76.67 | 79.65 |
| MaPLe [CVPR'23] | 80.82 | 78.70 | 79.75 | 80.36 | 59.18 | 68.16 | 94.07 | 73.23 | 82.35 | 83.00 | 78.66 | 80.77 |
| PromptSRC [ICCV'23] | 82.67 | 78.47 | 80.52 | 83.37 | 62.97 | 71.75 | 92.90 | 73.90 | 82.32 | 87.10 | 78.80 | 82.74 |
| MMA [CVPR'24] | 82.27 | 78.57 | 80.38 | 83.20 | 65.63 | 73.38 | 85.46 | 82.34 | 83.87 | 86.23 | 80.03 | 82.20 |
| DeKg [ICLR'25] | 82.52 | 78.30 | 80.35 | **83.80** | 59.66 | 69.70 | 94.02 | 81.69 | 87.42 | 88.06 | 81.77 | 84.80 |
| **MAIL** [Ours] | 82.50 | 78.70 | 80.56 | 83.15 | 67.39 | 74.45 | 93.50 | 85.11 | 89.11 | 87.34 | 80.22 | 83.63 |

utilizing only approximately 1/140 of ProS's training data, highlighting its remarkable efficiency in low-data scenarios. ❷ *Modality-coupled methods consistently outperform modality independent methods.* For instance, under FS-UCDR's UnseenGallery scenario, MaPLe and MMA achieve average $mAP_{200}$ improvements of 4.44% and 3.22% over IVLP and IVLA, respectively. This emphasizes the importance of collaboration and information sharing between modalities in low-data settings. ❸ *The vision-only methods perform similarly to, or even slightly outperform, the modality-independent methods.* As seen in the table, AdaptFormer achieves results comparable to IVLA, while VPT-Deep achieves an average mAP improvement of 0.5%-1.6% over IVLP. Therefore, we conclude that the benefit of simply fine-tuning the text side for retrieval is limited.

**Comparison Results under FS-U$^C$CDR**. In **Tab. 3**, we compare the FS-U$^C$CDR performance of our MAIL with other baselines. The results demonstrate that MAIL consistently achieves the best performance among all methods, indicating its effectiveness in enhancing CLIP's capability to handle semantic shifts under limited-data scenarios. Moreover, it can be observed that adapter-based methods and LoRA perform relatively poorly under the FS-U$^C$CDR setting.

**Comparison Results under Few-Shot Classification**. In **Tab. 4**, we compare the base-to-novel performance of MAIL against existing baselines across 11 datasets, reporting accuracies on base and novel classes, along with their harmonic mean (HM). Without relying on any regularization loss, MAIL achieves consistent gains of 0.23%, 1.01%, and 0.66% in Base, Novel, and HM, respectively, surpassing the previous best method, DeKg [30]. The improvement on novel classes is particularly notable, as DeKg [30] depends on regularization to enhance generalization, while MAIL attains better results with a simpler, regularization-free design. For results under the other two evaluation settings, please refer to **Tab. 5** and **Tab. 6**, where our MAIL also showcases strong performance.

### 5.4 Ablation Studies

In this section, we assess the performance of each component within MAIL. By default, we present the *average performance* scores for the UnseenGallery and MixedGallery scenarios on DomainNet across five query domains under FS-UCDR. More ablation studies can be found in **Appendix G, H**.

Table 5: Comparison of MAIL with previous state-of-the-art methods on cross-dataset evaluation.

| | Source | Target | | | | | | | | | | |
|---|---|---|---|---|---|---|---|---|---|---|---|---|
| | ImageNet | Average | Caltech101 | OxfordPets | StanfordCars | Flowers101 | Food101 | FGVCAircraft | SUN397 | DTD | EuroSAT | UCF101 |
| CoOp [IJCV'22] | 71.51 | 63.88 | 93.70 | 89.14 | 64.51 | 68.71 | 85.30 | 18.47 | 64.15 | 41.92 | 46.39 | 66.55 |
| CoOpOp [CVPR'22] | 71.02 | 65.74 | 94.43 | 90.14 | 65.32 | 71.88 | 86.06 | 22.94 | 67.36 | 45.73 | 45.37 | 68.21 |
| MaPLe [CVPR'23] | 70.72 | 66.30 | 93.53 | 90.49 | 65.57 | 72.23 | 86.20 | 24.74 | 67.01 | 46.49 | 48.06 | 68.69 |
| PromptSRC [ICCV'23] | 71.27 | 65.81 | 93.60 | 90.25 | 65.70 | 70.25 | 86.15 | 23.90 | 67.10 | 46.87 | 45.50 | 68.75 |
| MMA [CVPR'24] | 71.00 | 66.61 | 93.80 | 90.30 | 66.13 | 72.07 | 86.12 | 25.33 | 68.17 | 46.57 | 49.24 | 68.32 |
| DeKg [ICLR'25] | 72.33 | 66.64 | 94.73 | 90.02 | 65.49 | 72.39 | 86.59 | 25.05 | 67.19 | 44.47 | 51.37 | 68.78 |
| **MAIL** [Ours] | 72.10 | 67.02 | 94.73 | 91.37 | 66.63 | 71.47 | 86.33 | 25.27 | 67.30 | 45.47 | 52.80 | 68.87 |

Table 6: Comparison of MAIL with previous state-of-the-art methods on domain generalization across 4 datasets. These results of DeKg are derived from their OpenReview comment section.

| | Source | Target | | | | |
|---|---|---|---|---|---|---|
| | ImageNet | Average | -V2 | -S | -A | -R |
| CLIP [ICML'21] | 66.73 | 57.17 | 60.83 | 46.15 | 47.77 | 73.96 |
| CoOp [IJCV'22] | 71.51 | 59.27 | 64.20 | 47.99 | 49.71 | 75.21 |
| CoOpOp [CVPR'22] | 71.02 | 59.91 | 64.07 | 48.75 | 50.63 | 76.18 |
| MaPLe [CVPR'23] | 70.72 | 60.28 | 64.07 | 49.15 | 50.90 | 76.98 |
| PromptSRC [ICCV'23] | 71.27 | 60.65 | 64.35 | 49.55 | 50.90 | **77.80** |
| MMA [CVPR'24] | 71.00 | 60.47 | 64.33 | 49.13 | 51.12 | 77.32 |
| DeKg [ICLR'25] | 72.33 | 59.89 | 64.31 | 48.38 | 50.51 | 76.37 |
| **MAIL** [Ours] | 72.10 | 60.68 | 64.50 | 49.67 | 50.70 | **77.80** |

**Variants of AL and MAIL.** We first compare the performance of adding ALs to both encoders (IVLU) versus only to the image encoder (VU), as shown in the first two rows of **Tab. 7**, focusing exclusively on the image encoder yields better results. Next, we analyze the directionality of information flow in MAIL. Vision-to-language flow ($V \to L$) reduces the model's representational capacity, as evidenced by its poorer performance compared to IVLU. Similarly, the bidirectional flow ($V \leftrightarrow L$) underperforms compared to language-guided alignment ($L \leftarrow V$). This is likely due to visual features, which often include significant background noise and limited category-specific information, diluting the discriminability of text features. In contrast, text features, being semantically compact and category-specific, are better suited to guide alignment effectively without being adversely affected by redundant noisy information.

Table 7: Ablation studies on variants of AL and MAIL.

| Methods | UnseenGallery | | MixedGallery | | Params. (↓) |
|---|---|---|---|---|---|
| | mAP$_{200}$ | Prec$_{200}$ | mAP$_{all}$ | Prec$_{100}$ | |
| VU | 59.13 | 54.59 | 53.34 | 49.24 | **76288** |
| IVLU | 58.88 | 54.29 | 53.13 | 49.06 | 127488 |
| $V \to L$ | 58.67 | 54.00 | 52.97 | 48.79 | 637400 |
| $V \leftrightarrow L$ | 61.75 | 58.07 | 55.83 | 52.58 | 1147392 |
| **$V \leftarrow L$** | **62.05** | **58.31** | **56.05** | **52.80** | 637400 |

**Variants of Adding AL.** We begin by evaluating the effect of placing ALs at the final LN and projection layers of both encoders, as shown in **Fig. 4-a**. Results reveal a significant performance boost when ALs are applied to these locations. Please note that when we add AL to the projection layer, we also set the projection layer trainable. Next, we progressively distribute ALs across transformer blocks, from the first layer to the $l$-th block ($l = 1, 2, \ldots, 12$). As shown in **Fig. 4-b**, performance peaks when ALs are added to all 12 layers. Furthermore, we examine the effect of removing ALs from key transformer components—MLP, attention, and LN layers. **Fig. 4-c** shows that stepwise removal consistently degrades performance, highlighting the critical role of ALs across all components.

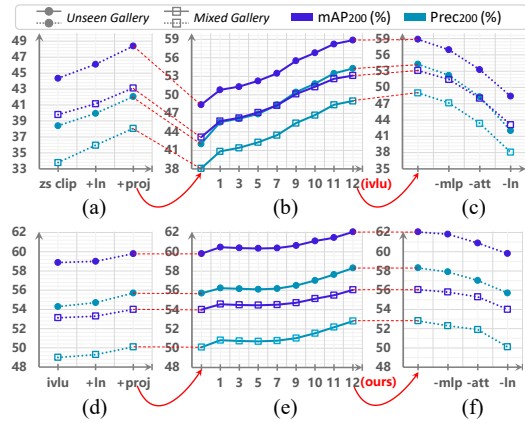

Figure 4: Ablation studies on different configurations of adding AL and MAIL.

**Variants of Adding MAIL.** Building on IVLU, we integrate the bottleneck-based bridge function into ALs, transforming them into MAILs. We first place the bridge function in the final LN and projection layers of both encoders, as illustrated in **Fig. 4-d**, which delivers promising results. Next, we incrementally add the bridge function to ALs in each transformer block, as shown in **Fig. 4-e**. Notably, introducing the bridge function in the first block yields a significant performance boost. While subsequent additions initially cause minor declines, a sharp improvement begins at the 10th block, culminating in the best performance when applied across all 12 blocks. Finally, we remove the bridge function (degrading MAIL to AL) systematically from each transformer block. **Fig. 4-f** reveals that removing it from the LN layer has the most pronounced negative effect.

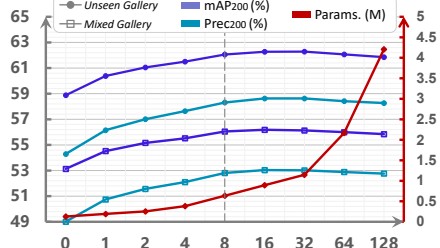
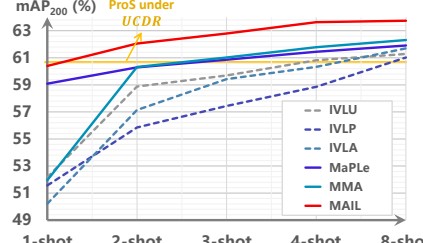

Figure 5: Ablation studies on different ranks. $rank\!=\!0$ denotes IVLU.

Figure 6: Results with different shots under FS-UCDR (UnseenGallery).

**Rank of the Bridge Function.** To evaluate the impact the rank of the bridge function in our MAIL, we conduct an ablation study by varying the rank systematically. As shown in **Fig. 5**, performance peaks when the rank is set to 16. However, increasing the rank beyond 16 leads to a slight decline in performance, likely due to the additional parameters increasing the risk of overfitting. To achieve a better trade-off, we select a rank of 8 for the final configuration.

**Results with Different Shots. Fig. 6** presents the results of our method compared to other methods across varying shot numbers, with detailed numerical values provided in **Appendix I**. This figure clearly demonstrates that the performance of all methods improves as the number of shots increases. Notably, our method consistently outperforms the its competitors. Additionally, it is worth noting that nearly all methods surpass ProS [13] with proper shots.

## 6 Conclusion and Limitation

In this paper, we introduce FS-UCDR, a practical setting that alleviates the data scarcity challenge in UCDR. Accordingly, we propose MAIL, a novel approach that enhances update alignment between modalities through coupled agent layers. Leveraging a scaling-and-shifting reparameterization mechanism, these agent layers are seamlessly integrated into the original CLIP, preserving inference efficiency while improving adaptability. Extensive experiments across three benchmarks validate its effectiveness. Beyond retrieval, MAIL's alignment strategy also holds promise for few-shot classification.

The **limitation** of MAIL lies in: MAIL's sequential nature, i.e., $\text{AL} \circ \text{OP}(x)$ leads to slight longer training time and memory (as we have provided in the **Appendix E**) compared with MaPLe and MMA. We will explore optimizations in future work.

## 7 Acknowledgments and Disclosure of Funding

This work was supported by the National Natural Science Foundation of China (Nos. 62476056 and 62306070) and the Social Development Science and Technology Project of Jiangsu Province (No. BE2022811). This work was also supported in part by the Southeast University Start-Up Grant for New Faculty under Grant 4009002309. Furthermore, the work was supported by the Big Data Computing Center of Southeast University and the SEU Innovation Capability Enhancement Plan for Doctoral Student (CXJH_SEU 25133). This work was also supported by "the Fundamental Research Funds for the Central Universities (2242025K30024)".

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

# Summary of the Appendix

In the appendix of this paper, we provide further details:

- Elaboration on the used datasets (Appendix A).

- Explanation on the used metrics (Appendix B).

- Explanation on the implementation details (Appendix C).

- Elaboration on the used loss functions (Appendix D).

- Elaboration on the computational cost (Appendix E).

- Elaboration on the implementations for baseline methods (Appendix F).

- Additional design choices and details (Appendix G).

- Additional visualization results (Appendix H).

- Detailed experiment results with other shots under FS-UCDR (Appendix I).

## A  Datasets

### A.1  Datasets for FS-UCDR and Its Variants

We conduct experiments on three datasets: DomainNet [38], Sketchy [41, 32], and TU-Berlin [12, 53]. **DomainNet** is utilized for UCDR and $U^D$CDR evaluations, comprising 596,006 images across six domains: *Real*, *Sketch*, *Quickdraw*, *Infograph*, *Clipart*, and *Painting*. Following the leave-one-out protocol from ProS, five domains serve as sources, while the remaining one acts as the unseen query domain. The $\mathrm{MixedGallery}$ is created by combining the $\mathrm{UnseenGallery}$ with 8% of samples from each seen class in the *Real* domain. For $U^D$CDR evaluation, we select 45 training classes and use 25% of the samples from each class for both the query domain (10% for *Quickdraw*) and the *Real* domain. The **Sketchy** and **TU-Berlin** datasets are employed for $U^C$CDR evaluation, each containing two domains: *Real* and *Sketch*. Detailed statistics of the datasets are summarized in **Tab. 8**.

Table 8: Statistics of the utilized datasets for FS-UCDR and its variants. The average shots denotes the average number of images per class in each domain.

| Dataset | Images | Domains | Classes | Train Classes | Val Classes | Test Classes | Average Shots |
|---|---|---|---|---|---|---|---|
| DomainNet | 596006 | 6 | 345 | 245 | 55 | 45 | 287.9 |
| Sketchy | 148473 | 2 | 125 | 93 | 11 | 21 | 593.9 |
| TU-Berlin | 224489 | 2 | 250 | 200 | 20 | 30 | 449.0 |

### A.2  Datasets for Few-Shot Classification.

**Base-to-Novel Generalization**: In this evaluation, the dataset's categories are partitioned equally into base and novel classes. The model is trained solely on the base classes and evaluated on both base and novel classes. This setup enables us to assess the model's transfer learning performance on seen categories. We conduct this evaluation across 11 diverse image classification datasets: ImageNet [9], Caltech101 [14], OxfordPets [36], StanfordCars [29], Flowers102 [34], Food101 [2], FGVCAircraft [33], SUN397 [48], UCF101 [35], DTD [8], and EuroSAT [17].

**Cross-Dataset Evaluation:** This evaluation tests how well the model works on new datasets it has never seen before. Like CoCoOp [56], we first train the model on all 1000 ImageNet classes using only a few examples per class. Then, we directly apply the trained model to other datasets to see if it can generalize across datasets. The target datasets used are the ten remaining datasets in the base-to-novel generalization experiment.

**Domain Generalization:** In this setup, similar to cross-dataset evaluation, we also use ImageNet for training, and evaluate on four domain-shifted variants—ImageNetV2 [40], ImageNet-Sketch [46], ImageNet-A [19], and ImageNet-R [18]—each presenting a distinct type of domain variation.

Detailed statistics of the datasets are summarized in **Tab. 9**.

Table 9: Statistics of the utilized datasets for few shot classification. * denotes the number of images.

| Dataset | Classes | Train* | Val* | Test* | Prompt |
|---|---|---|---|---|---|
| ImageNet | 1000 | 1.28M | ∼ | 50000 | "a photo of a <category>." |
| Caltech101 | 100 | 4128 | 1649 | 2465 | "a drawing of a <category>." |
| OxfordPets | 37 | 2944 | 736 | 3669 | "an awesome animal pet photo of a <category>." |
| StanfordCars | 196 | 6509 | 1635 | 8041 | "a photo of my <category>." |
| Flowers102 | 102 | 4093 | 1633 | 2463 | "a flower photo of a <category>." |
| Food101 | 101 | 50500 | 20200 | 30300 | "a food photo of a <category>." |
| FGVCAircraft | 100 | 3334 | 3333 | 3333 | "a brand aircraft a <category>." |
| SUN397 | 397 | 15880 | 3970 | 19850 | "a scene photo of a <category>." |
| DTD | 47 | 2820 | 1128 | 1692 | "a beautiful texture drawing a <category>." |
| EuroSAT | 10 | 13500 | 5400 | 8100 | "a photo of a <category>, a type of centered satellite." |
| UCF101 | 101 | 7639 | 1898 | 3783 | "a photo of a <category>, a type of action." |
| ImageNetV2 | 1,000 | ∼ | ∼ | 10,000 | "a photo of a <category>." |
| ImageNet-Sketch | 1,000 | ∼ | ∼ | 50,889 | "a sketch photo of a <category>." |
| ImageNet-A | 200 | ∼ | ∼ | 7,500 | "a poor photo of a <category>." |
| ImageNet-R | 200 | ∼ | ∼ | 30,000 | "a sketch photo of a <category>." |

## A.3 Practicality of FS-UCDR

**1. Data collection is hard.** UCDR relies on DomainNet [38] as its benchmark dataset, where each domain contains an average of 287.9 samples per class, resulting in approximately 1,400 samples per class across five diverse domains. For comparison, ImageNet—commonly used in few-shot learning—contains roughly 1,200 to 1,300 samples per class. Although the per-class sample sizes are similar, collecting DomainNet-style multi-domain data is significantly more challenging than collecting single-domain (real-world) images like those in ImageNet. While automated data collection may be feasible for domains such as Real, it is much harder for others like Clipart or Sketch. Therefore, developing and studying the few-shot UCDR setting is both meaningful and necessary, as it reduces data requirements while maintaining the core challenge of cross-domain generalization.

**2. More data does not help.** Since UCDR involves unseen domains and classes during testing, the inherent domain and semantic shifts between training and testing phases may increase the risk of overfitting for PEFT methods when more training data is used. This phenomenon is supported by empirical evidence: as shown in the **Tab. 10**, the average mAP for 2-shot and full-shot settings is 62.05% and 61.95%, respectively. This indicates that increasing the number of training samples does not necessarily lead to better performance, further highlighting the relevance of the FS-UCDR setting.

Table 10: The average performance of MAIL across five query domains under varying shot settings in the FS-UCDR task. Bold values denote the best performance. Results from 2-shot UCDR experiments are highlighted with a light purple background, while those from full-shot UCDR experiments are highlighted with a light grey background.

| Shot | UnseenGallery | | MixedGallery | |
|---|---|---|---|---|
| | $mAP_{200}$ | $Prec_{200}$ | $mAP_{all}$ | $Prec_{100}$ |
| 1 | 60.40 | 56.12 | 54.70 | 50.87 |
| 2 | 62.05 | 58.31 | 56.05 | 52.80 |
| 4 | 63.64 | 59.94 | 57.71 | 54.63 |
| 8 | 63.75 | 60.19 | 57.72 | 54.62 |
| 16 | **63.77** | **60.42** | **57.75** | **54.66** |
| 32 | 63.52 | 60.13 | 57.52 | 54.13 |
| 64 | 63.21 | 59.82 | 57.31 | 53.95 |
| 128 | 62.79 | 59.42 | 57.01 | 53.54 |
| FULL (287.9) | 61.95 | 58.12 | 56.60 | 52.96 |

# B Evaluation Metrics

For FS-UCDR, FS-U$^D$CDR, and FS-U$^C$CDR, following prior work ProS [13], we adopt the same evaluation metrics. For the Sketchy and DomainNet datasets, **precision** (Prec$_{200}$) and **mean Average Precision** (mAP$_{200}$) are calculated based on the top-200 retrieved results. For the TU-Berlin dataset, we use Prec$_{100}$ and mAP$_{all}$ as the evaluation metrics. As for the few-shot classification, the **classification accuracy** is adopted.

# C Implementation Details.

## C.1 Implementation Details for FS-UCDR and Its Variants

When CLIP [39] is applied in UCDR tasks, the ViT-B/32 [11] backbone is most commonly used [13]. We follow this convention and utilize a pre-trained ViT-B/32 [11] CLIP model with $d_t = 512$ and $d_v = 768$, the rank ($r$) of the bridge function in MAIL, is set to 8. The text prompt is fixed as "a photo of a <category>".

For DomainNet, training is limited to 1 epoch, whereas Sketchy and TU-Berlin are trained for 20 epochs, with early stopping applied after 2 epochs. Given the diverse source domains in FS-UCDR, we organize each batch as $B = N_s \times C_b \times k_b$, where $N_s$ represents the number of source domains, $C_b = 3$ denotes the number of classes sampled from each domain (we sample different classes for each domain), and $k_b = 4$ denotes the number of images for each class from each source domain within the batch. Note that if $k_b > k$, repeated images will appear in the batch, with $k$ denotes the shot number.

The optimization is performed using the Adam optimizer with a learning rate of $2e-4$ and a cosine decay schedule. All experiments are conducted on a single NVIDIA RTX 4090 GPU with mixed-precision training to accelerate computation. To ensure reproducibility, we follow the setting in ProS [13] and fix the random seed to 0. A 2-shot training strategy is employed, where two samples per class per domain are randomly selected. For the used loss functions and the results with other shot configurations, one can refer to **Appendix D** and **Appendix I**.

## C.2 Implementation Details for Few-Shot Classification

We follow prior studies [57, 56, 24, 49], the ViT-B/16 [11] variant of the CLIP model serves as the visual backbone for all experimental setups, with $d_t = 512$ and $d_v = 768$, the rank ($r$) of the bridge function in MAIL, is set to 32. Hand-crafted text prompts from prior methods [39, 57, 55] are utilized and described in detail in **Tab. 9**. A 16-shot training strategy is employed, where 16 samples per class are randomly selected. The average accuracy is reported over three independent runs with random seeds set to 0, 1, and 2. All experiments are conducted on a single NVIDIA RTX 4090 GPU.

For **base-to-novel evaluation**, we adopt a batch size of 64 for the larger datasets (ImageNet and SUN397) and 4 for all others. Training is performed for 5 epochs on ImageNet and 10 epochs on the remaining datasets. We employ the AdamW optimizer for all experiments, except on EuroSAT, where the SGD optimizer yields better performance. The initial learning rate is set to $5.0 \times 10^{-6}$ for Food101, $2.5 \times 10^{-5}$ for DTD, and $1.5 \times 10^{-5}$ for the other datasets. The rank ($r$) of the bridge is set to 12 for DTD and 32 for the remaining datasets.

For **cross-dataset evaluation and domain generalization tasks,** we train the model on ImageNet for 2 epochs. Due to GPU memory constraints—mainly caused by the full 1000-class setting, which is twice the size of the base-to-novel evaluation (500 classes)—we reduce the batch size from 64 to 32 and use half-precision training (fp16). The initial learning rate is set to $2.5e-5$ .

# D Loss Function

Given the diverse source domains in FS-UCDR, we organize each batch as $B = N_s \times C_b \times k_b$, where $N_s$ represents the number of source domains, $C_b = 3$ denotes the number of classes sampled from each domain (we sample different classes for each domain), and $k_b = 4$ denotes the number of images for each class from each source domain within the batch. Note that if $k_b > k$, repeated images will appear in the batch, with $k$ denotes the shot number. We utilize two loss functions: the image-text matching loss and the triplet-hard loss [20]. The image-text matching loss is defined as a

cross-entropy loss:

$$\mathcal{L}_{ce} = \frac{1}{B} \sum_{j=1}^{B} -y_j \log \frac{\exp(sim(\mathcal{T}(t_{y_j}), \mathcal{V}(I_j)))}{\sum_{i=1}^{C} \exp(sim(\mathcal{T}(t_i), \mathcal{V}(I_j)))}, \quad (9)$$

where $I_j$ is the $j$-th image in the batch, and $y_j$ is the corresponding label.

The triple-hard loss is formulate as:

$$\mathcal{L}_{tri\_hard} = \frac{1}{B} \sum_{i=1}^{P} \sum_{a=1}^{K} [\rho - \min_{p=1...K} sim(\mathcal{V}(I_i^a), \mathcal{V}(I_i^p)) + \max_{\substack{j=1...P \\ j \neq i \\ n=1...K}} sim(\mathcal{V}(I_i^a), \mathcal{V}(I_j^n))]_+, \quad (10)$$

where $\rho = 0.5$ is the margin hyper-parameter, $P = C_b$ denotes the number of identities within the batch, and $K = N_S \times k_b$ is the total number of samples for each class across all the source domains. $[\cdot]_+$ denotes the $max(0, \cdot)$ function, $sim(\cdot, \cdot)$ represents cosine similarity, and $I_i^a$ identifies the anchor image, specifically the $a$-th image from the $i$-th class within the batch. Additionally, $I_i^p$ refers to the positive sample, while $I_j^n$ corresponds to the negative sample.

In terms of the **few-classification task**, we only utilize the cross-entropy loss.

# E   Computational Cost

To validate the efficiency of our method, we present the computational costs of MAIL in **Fig. 7**. Both the training and inference stages utilize a batch size of 60. Compared to MMA and MaPLe, while MAIL incurs higher computational costs during training—including memory usage and training time—it reduces test time, inference memory, and GFLOPs. These results highlight the superior efficiency of MAIL during inference. Please note these results are obtained under FS-UCDR's UnseenGallery scenario, where *Sketch* domain is used as the query domain.

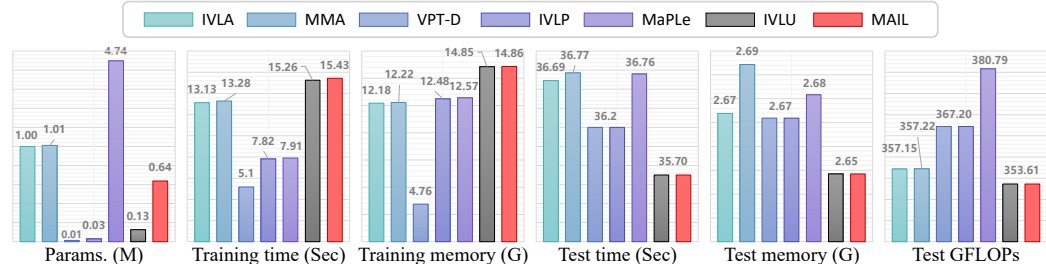

Figure 7: Computational costs of various tuning methods, presented from left to right: the number of trainable parameters, training time (per epoch), training memory usage, test time, test memory usage, and test GFLOPs.

# F   Implementation of Baseline Methods under FS-UCDR and Its Variants

We re-implement VPT-Deep [23], IVLP [24], MaPLe [24], MMA [49], AdaptFormer [6] and IVLA [49] based on the released code. Specifically, the configurations for VPT-Deep, IVLP, and MaPLe are mainly taken directly from the MaPLe paper. However, while the original prompt depth for MaPLe ranges from 1 to 9, we find that a depth of 1 to 12 performs significantly better for our task. The configurations for VPT-Deep, IVLP and MaPLe, are as follows:

Table 11: The used configurations for VPT-D, IVLP, and MaPLe. PL denotes the prompt length.

| METHOD | PL-VISUAL | PL-TEXTUAL | PROMPT DEPTH | LEARNING RATE |
|--------|-----------|------------|--------------|---------------|
| VPT-D  | 4         | 0          | 1-12         | 0.0002        |
| IVLP   | 2         | 2          | 1-12         | 0.0002        |
| MAPLE  | 2         | 2          | 1-12         | 0.0002        |

For AdaptFormer, IVLA and MMA, we adopt the configurations outlined in the MMA paper, with one exception regarding the depth parameter. While the MMA paper sets the depth to 9-12 for cross-dataset evaluation, we find that a depth of 1-12 performs better under FS-UCDR. The configurations for AdaptFormer, IVLA and MMA, are as follows:

Table 12: The used configurations for IVLA and MMA.

| METHOD | ADAPTER RANK | ADAPTER DEPTH | LEARNING RATE |
|---|---|---|---|
| ADAPTFORMER | 32 | 1-12 | 0.0015 |
| IVLA | 32 | 1-12 | 0.0015 |
| MMA | 32 | 1-12 | 0.0015 |

For LoRA[22], we adopt the implementation from CLIP-LoRA [22], with an initial learning rate of 0.0005, and the rank of the low-rank matrices, is set to 6 after hyper-parameter tuning. For BitFit [52], we fine-tune **all** bias terms in the CLIP backbone, using an initial learning rate of 0.0002. All other configurations, including batch size and loss functions, are kept identical to those used in our MAIL.

Note that LoRA performs poorly from the main body. To further investigate this issue, we conduct additional experiments under the FS UCDR setting, where we replace CLIP with our trained MAIL as the backbone for LoRA. The results are presented in **Tab. 13**, showing that LoRA remains inferior to MAIL and even degrades its performance. We believe this is an interesting open question for future research—why LoRA, despite its effectiveness in many scenarios, performs poorly in this context.

Table 13: FS-UCDR (2-shot) evaluation results (%) on DomainNet. * denotes the results are obtained using the full training data, i.e., the UCDR results. $^{\dagger}$ denotes that the results are obtained when our MAIL is used as backbone.

| Methods | Sketch | | | | Quickdraw | | | | Painting | | | |
|---|---|---|---|---|---|---|---|---|---|---|---|---|
| | UnseenGallery | | MixedGallery | | UnseenGallery | | MixedGallery | | UnseenGallery | | MixedGallery | |
| | mAP$_{200}$ | Prec$_{200}$ | mAP$_{200}$ | Prec$_{200}$ | mAP$_{200}$ | Prec$_{200}$ | mAP$_{200}$ | Prec$_{200}$ | mAP$_{200}$ | Prec$_{200}$ | mAP$_{200}$ | Prec$_{200}$ |
| ProS* [CVPR'24] | 64.67 | 60.01 | 58.43 | 54.63 | 28.42 | 25.44 | 23.18 | 21.27 | 75.16 | 69.55 | 71.20 | 66.12 |
| CLIP [ICML'21] | 42.20 | 35.28 | 36.62 | 29.79 | 7.44 | 5.61 | 6.00 | 3.17 | 61.68 | 55.07 | 56.53 | 50.14 |
| LoRA [ICLR'22] | 54.85 | 49.23 | 48.71 | 43.33 | 22.16 | 18.10 | 17.73 | 14.21 | 71.46 | 65.10 | 66.81 | 60.81 |
| LoRA$^{\dagger}$ [ICLR'22] | 64.74 | 60.41 | 56.82 | 52.90 | 27.98 | 25.39 | 20.40 | 18.63 | 74.95 | 68.37 | 69.44 | 64.33 |
| **MAIL** [Ours] | **65.76** | **61.57** | **59.05** | **55.25** | **29.41** | **26.95** | **22.83** | 21.26 | **76.05** | **70.85** | **71.12** | **66.44** |

| Methods | Infograph | | | | Clipart | | | | Average | | | |
|---|---|---|---|---|---|---|---|---|---|---|---|---|
| | UnseenGallery | | MixedGallery | | UnseenGallery | | MixedGallery | | UnseenGallery | | MixedGallery | |
| | mAP$_{200}$ | Prec$_{200}$ | mAP$_{200}$ | Prec$_{200}$ | mAP$_{200}$ | Prec$_{200}$ | mAP$_{200}$ | Prec$_{200}$ | mAP$_{200}$ | Prec$_{200}$ | mAP$_{200}$ | Prec$_{200}$ |
| ProS* [CVPR'24] | 57.98 | 54.42 | 52.19 | 49.56 | 76.48 | 71.86 | 72.28 | 68.15 | 60.52 | 56.26 | 55.46 | 51.95 |
| CLIP [ICML'21] | 50.08 | 44.74 | 43.75 | 38.91 | 60.37 | 51.30 | 56.08 | 46.91 | 44.35 | 38.40 | 39.80 | 33.78 |
| LoRA [ICLR'22] | 58.01 | 53.84 | 51.86 | 48.20 | 70.52 | 64.11 | 66.00 | 59.63 | 55.40 | 50.08 | 50.22 | 45.24 |
| LoRA$^{\dagger}$ [ICLR'22] | 58.96 | 55.04 | 52.30 | 50.26 | 77.12 | 72.56 | 72.02 | 67.25 | 60.75 | 56.35 | 54.20 | 50.67 |
| **MAIL** [Ours] | **60.11** | **57.40** | **53.34** | **50.95** | **78.94** | **74.80** | **73.91** | **70.14** | **62.05** | **58.31** | **56.05** | **52.80** |

# G  Additional Design Choices and Details

## G.1  Linear Layer Bridge Function

The bridge function in our MAIL is a bottleneck structure, although it could be implemented as a simple linear layer. In our experiments, we find that the linear layer not only has significantly more trainable parameters but also performs worse than the bottleneck structure. The results are as follows:

Table 14: Ablation studies on linear layer bridge function. We present the average performance scores on DomainNet across five query domains under FS-UCDR.

| Methods | UnseenGallery | | MixedGallery | | Params. ($\downarrow$) |
|---|---|---|---|---|---|
| | mAP$_{200}$ | Prec$_{200}$ | mAP$_{all}$ | Prec$_{100}$ | |
| V $\rightarrow$ L (Linear Layer) | 58.49 | 54.65 | 53.53 | 49.50 | 19657216 |
| V $\leftrightarrow$ L (Linear Layer) | 61.32 | 57.46 | 55.39 | 52.04 | 39186944 |
| V $\leftarrow$ L (Linear Layer) | 61.40 | 57.53 | 55.52 | 52.16 | 19657216 |
| **V $\leftarrow$ L (Bottleneck)** | **62.05** | **58.31** | **56.05** | **52.80** | **637400** |

## G.2 Initialization Method of the Bridge Function

Next, we exam the initialization method of the bridge function. As noted in the main body, $W_{down}$ is initialized with random Gaussian values, while $W_{up}$ is initialized to zeros, i.e., $W_{down} \sim \mathcal{N}(0, \sigma_t^2)$, with $\sigma_t = \frac{1}{\sqrt{d_t}}$, $W_{up} = 0$. Here we conduct more experiments about the initialization of the two matrices, the results, are as follows:

Table 15: Ablation studies on different initialization methods of $W_{down}$ and $W_{up}$. Note that if $W_{down}$ or $W_{up}$ are initialized with Uniform values, then, $W_{down} \sim U(-\frac{\sqrt{12\sigma_t^2}}{2}, \frac{\sqrt{12\sigma_t^2}}{2})$ and $W_{up} \sim U(-\frac{\sqrt{12\sigma_v^2}}{2}, \frac{\sqrt{12\sigma_v^2}}{2})$, with $\sigma_v = \frac{1}{\sqrt{d_v}}$. Similarly, if $W_{up}$ is initialized with Gaussian values, then, $W_{up} \sim \mathcal{N}(0, \sigma_v^2)$. We present the average performance scores for the UnseenGallery and MixedGallery scenarios on DomainNet across five query domains under FS-UCDR.

| $W_{down}$ | $W_{up}$ | UnseenGallery | | MixedGallery | |
|---|---|---|---|---|---|
| | | mAP$_{200}$ | Prec$_{200}$ | mAP$_{all}$ | Prec$_{100}$ |
| 0 | Uniform | 59.94 | 56.05 | 53.99 | 50.53 |
| 0 | Gaussian | 60.47 | 56.48 | 54.48 | 50.98 |
| 0 | 0 | 58.77 | 54.17 | 53.08 | 48.92 |
| Uniform | Uniform | 59.61 | 55.65 | 53.66 | 50.14 |
| Uniform | 0 | 62.01 | 58.16 | 55.83 | 52.59 |
| Gaussian | Gaussian | 59.89 | 55.88 | 53.95 | 50.43 |
| Gaussian | 0 | **62.05** | **58.31** | **56.05** | **52.80** |

From **Tab. 15**, we observe that the Gaussian distribution achieves slightly better performance compared with the uniform distribution. We encourage readers to explore additional distributions.

## G.3 Should the Shifting Vectors in the Agent Layers Be Connected Across Modalities?

Note that we only apply the bridge function to the scaling vectors $a$ across modalities. To address the question above, we conducted the following experiments, selectively adding or removing bridge functions from the shifting vectors $b$:

Table 16: Ablation studies on bridge function applied to the shifting vector $b$. We present the average performance scores on DomainNet across five query domains under FS-UCDR.

| Methods | UnseenGallery | | MixedGallery | | Params. ($\downarrow$) |
|---|---|---|---|---|---|
| | mAP$_{200}$ | Prec$_{200}$ | mAP$_{all}$ | Prec$_{100}$ | |
| $w$ Bridge Function | **62.26** | **58.50** | **56.23** | **53.01** | 1147392 |
| $w/o$ Bridge Function | 62.05 | 58.31 | 56.05 | 52.80 | **637400** |

As we can see from **Tab. 16**, the introduction of the bridge function to the shifting vectors $b$ does improve performance; however, this improvement is limited. Furthermore, it results in doubling the number of trainable parameters, which creates a poor trade-off. Therefore, we propose to remove the bridge function from the shifting vectors.

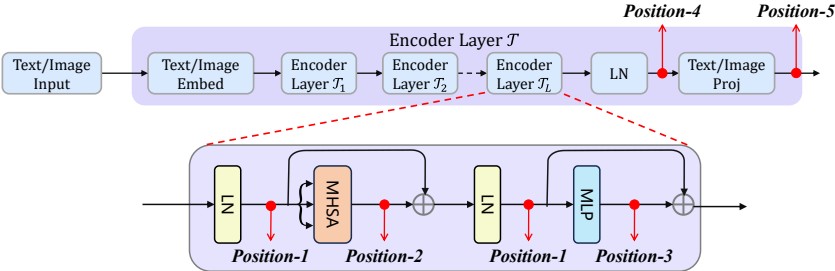

Figure 8: The five types of insertion locations.

## G.4 Discussion About the Insert Locations.

We have utilized about five types of insert locations (**Fig. 8**):

- **_Position-1&2&3_**: In the transformer block, insert the agent layer after each LayerNorm (LN) & MHSA & MLP layer.

- **_Position-4&5_**: Out of the transformer block, insert the agent layer after the last LN & projection layer.

We select these five positions for ease of implementation. For clarity, we illustrate only positions 1, 2, and 3. The pseudocode for the Transformer block is presented in **Alg. 1**. As shown, incorporating our modifications requires adding just four lines of code, as depicted in **Alg. 2**, ensuring minimal code changes. Additionally, the definition of MAIL is located outside the transformer block (**Alg. 3**).

---

**Algorithm 1** Pseudocode for the Transformer block within CLIP, presented in a PyTorch-like style.

```
# x:      features from the last transformer block
# y:      features to the next transformer block
# LN:     layer normalization
# MHSA:   multi-head self-attention
# MLP:    multilayer perceptron
# flag:   indicates whether the current block from the image encoder or the text encoder.

x = LN(x) # We can insert MAIL here
# 1. x = MAIL(x, flag)
mhsa = MHSA(x) # We can insert MAIL here
# 2. mhsa = MAIL(mhsa, flag)
x = x + mhsa
x = LN(x)# We can insert MAIL here
# 3. x = MAIL(x, flag)
mlp = MLP(x)# We can insert MAIL here
# 4. mlp = MAIL(mlp, flag)
y = x + mlp
```

---

**Algorithm 2** Pseudocode for the Transformer block with MAIL integration within CLIP, presented in a PyTorch-like style.

```
# flag:   indicates whether the current block from the image encoder or the text encoder.
# MAILs:  a list containing 4 MAIL blocks

x = LN(x)
x = MAILs[0](x, flag) # position-1
mhsa = MHSA(x)
mhsa = MAILs[1](mhsa, flag) # position-2
x = x + mhsa
x = LN(x)
x = MAILs[2](x, flag) # position-1
mlp = MLP(x)
mlp = MAILs[3](mlp, flag) # position-3
y = x + mlp
```

---

**Algorithm 3** Pseudocode of MAIL in a PyTorch-like style.

```
# x:          input feature
# y:          output feature
# flag:       indicates whether  x  from the image encoder or the text encoder.
# imageAgent: the agent layer for the image encoder
# textAgent:  the agent layer for the text encoder
# W:          bridge function

if flag=="text":
    y = x * textAgent.a + textAgent.b
else:
    a = imageAgent.a + textAgent.a @ W.down @ W.up
    y = x * a + visionAgent.b
```

---

# H  Visualization

## H.1  T-SNE Visualization

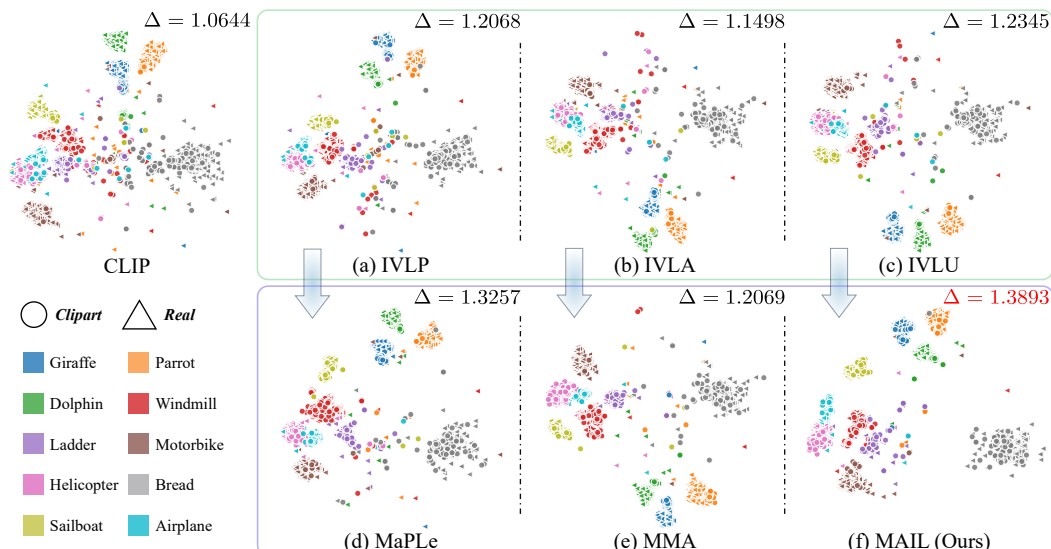

Figure 9: The t-SNE visualization for 10 randomly selected unseen classes of *Clipart* (query) domains and *Real* (gallery) domain. Different colors represent different classes, while $\triangle$ and $\bigcirc$ represent samples from *Real* and *Clipart* domains, respectively. we also evaluate the inter-class distinctiveness and intra-class compactness of feature space by the metric $\Delta = \frac{\text{mean } \mathcal{D}_{inter}}{\text{mean } \mathcal{D}_{intra}}$ (higher is better).

As shown in **Fig. 9**, we visualize the image features extracted from frozen CLIP, IVLP, IVLA, IVLU, MaPLe, MMA, and our MAIL method for 10 randomly selected unseen classes from the *Real* and unseen *Clipart* domains using t-SNE [44]. We also evaluate the inter-class distinctiveness and intra-class compactness of the feature space using the metric

$$\Delta = \frac{\text{mean } \mathcal{D}_{inter}}{\text{mean } \mathcal{D}_{intra}}, \tag{11}$$

where $\mathcal{D}_{inter}$ is a set that measures distances between all centers testing classes:

$$\mathcal{D}_{inter} = \{d(c^i, c^j) | i = 1, 2, ..., C_{test}, j = i + 1, ..., C_{test}\}. \tag{12}$$

Here, $C_{test}$ denotes the number of classes during the test, $c^i = \frac{1}{n_i} \sum_{k=1}^{n_i} b_k^i$ is the centroid of the $i$-th class, with $n_i$ being the number of samples in the $i$-th class, $b_k^i$ denoting the sample, and $d(\cdot, \cdot)$ representing the Euclidean distance.

Similarly, $\mathcal{D}_{intra}$ measures the distances among samples of the same class relative to their corresponding centroid:

$$\mathcal{D}_{intra} = \{d^i | i = 1, 2, ..., C_{test}\}, \tag{13}$$

where $d^i = \frac{1}{n_i} \sum_{k=1}^{n_i} d(b_k^i, c^i)$ indicates the compactness of the $i$-th class.

From **Fig. 9**, we can observe that modality-coupled methods better align feature representations of the same classes between the two domains and exhibit greater separation between different classes compared to modality-independent methods. Furthermore, our MAIL method achieves the best results in both the visualization and numeric metric evaluations.

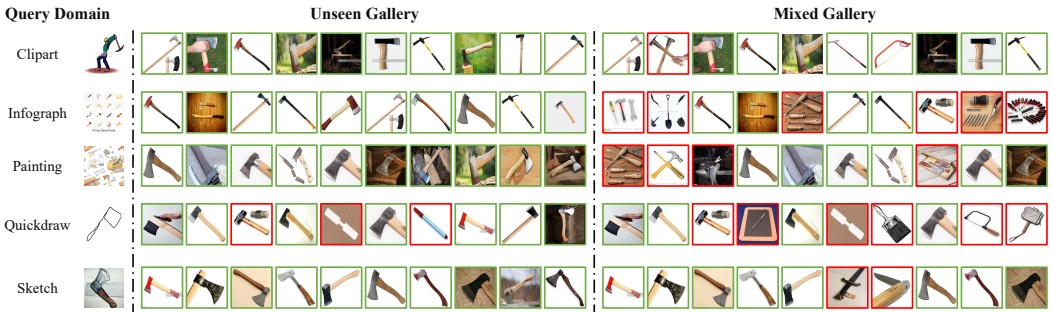

Figure 10: Retrieval results with query classes *axe*. Please zoom in for a better resolution.

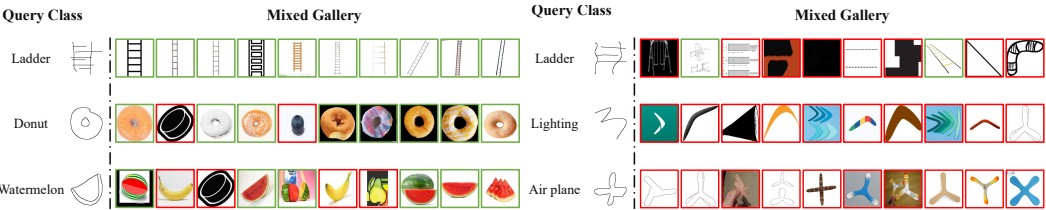

Figure 11: Retrieval results in query domain *Quickdraw*. Please zoom in for a better resolution.

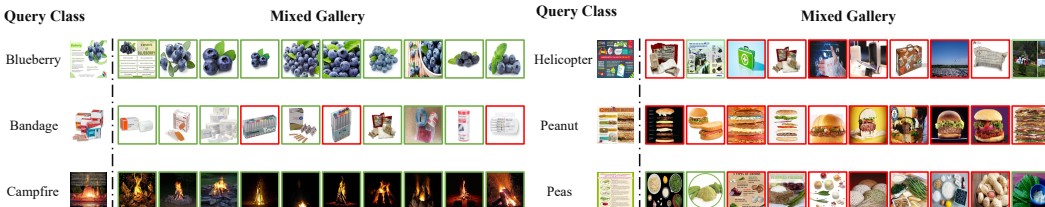

Figure 12: Retrieval results in query domain *Infograph*. Please zoom in for a better resolution.

## H.2 Top-10 Retrieved Results

To further evaluate the effectiveness of MAIL, we visualize several retrieval results. **Fig. 10** presents the top-10 retrieved candidates for the category "axe" using query images from different domains, while **Fig. 11** and **Fig. 12** display the top-10 retrieval results for queries originating from the *Quickdraw* domain.

These visualizations demonstrate that while MAIL achieves strong retrieval performance overall, its effectiveness is occasionally hindered by *query ambiguity*. This issue arises primarily from two factors: (1) **Lack of detail**: Queries from the *Quickdraw* domain often contain simplistic or abstract sketches, making them inherently ambiguous and prone to misinterpretation. (2) **Object clutter**: Queries from the *Infograph* domain frequently depict multiple objects within a single image, introducing uncertainty regarding the intended retrieval target. These challenges highlight potential areas for further refinement in handling ambiguous queries across diverse domains.

# I Evaluation Results with Other Shots

While the 2-shot results are presented in the main body, we provide additional results for other shot settings in this section: 1-shot, 3-shot, 4-shot, and 8-shot. As observed in the tables below, the performance of all methods improves with an increased number of shots. Notably, our method consistently outperforms its competitors. Furthermore, it is important to emphasize that nearly all methods surpass ProS with a sufficient number of shots (significantly fewer than full shots).

## I.1 1-Shot Results

Table 17: 1-shot UCDR evaluation results (%) on DomainNet.

| Methods | Sketch | | | | Quickdraw | | | | Painting | | | |
| | Unseen Gallery | | MixedGallery | | UnseenGallery | | MixedGallery | | UnseenGallery | | MixedGallery | |
| | $mAP_{200}$ | $Prec_{200}$ | $mAP_{200}$ | $Prec_{200}$ | $mAP_{200}$ | $Prec_{200}$ | $mAP_{200}$ | $Prec_{200}$ | $mAP_{200}$ | $Prec_{200}$ | $mAP_{200}$ | $Prec_{200}$ |
|---|---|---|---|---|---|---|---|---|---|---|---|---|
| ProS* | 64.67 | 60.01 | 58.43 | 54.63 | 28.42 | 25.44 | 23.18 | 21.27 | 75.16 | 69.55 | 71.20 | 66.12 |
| ZS CLIP | 42.20 | 35.28 | 36.62 | 29.79 | 7.44 | 5.61 | 6.00 | 3.17 | 61.68 | 55.07 | 56.53 | 50.14 |
| VPT-D | 55.16 | 49.98 | 48.48 | 43.71 | 21.22 | 19.20 | 15.94 | 14.36 | 69.71 | 63.05 | 64.76 | 58.46 |
| AdaptFormer | 49.73 | 43.15 | 43.55 | 37.33 | 11.35 | 9.29 | 8.63 | 5.88 | 65.00 | 57.75 | 59.86 | 53.04 |
| IVLP | 51.41 | 46.31 | 45.18 | 40.19 | 13.10 | 11.95 | 9.65 | 8.08 | 68.42 | 62.08 | 63.32 | 57.38 |
| IVLA | 51.23 | 44.82 | 44.94 | 38.93 | 12.72 | 10.58 | 9.69 | 6.90 | 65.75 | 58.61 | 60.63 | 53.92 |
| MaPLe | 60.31 | 55.51 | 53.59 | 49.20 | 25.91 | 23.52 | 20.06 | 18.29 | 74.03 | 68.18 | 69.27 | 63.74 |
| MMA | 53.15 | 46.85 | 46.15 | 40.83 | 15.29 | 12.68 | 11.66 | 8.75 | 66.73 | 59.81 | 61.63 | 55.15 |
| **MAIL** | **63.25** | **58.59** | **56.67** | **52.43** | **27.56** | **24.82** | **21.73** | **19.78** | **74.53** | **68.78** | **69.75** | **64.41** |

| Methods | Infograph | | | | Clipart | | | | Average | | | |
| | UnseenGallery | | MixedGallery | | UnseenGallery | | MixedGallery | | UnseenGallery | | MixedGallery | |
| | $mAP_{200}$ | $Prec_{200}$ | $mAP_{200}$ | $Prec_{200}$ | $mAP_{200}$ | $Prec_{200}$ | $mAP_{200}$ | $Prec_{200}$ | $mAP_{200}$ | $Prec_{200}$ | $mAP_{200}$ | $Prec_{200}$ |
|---|---|---|---|---|---|---|---|---|---|---|---|---|
| ProS* | 57.98 | 54.42 | 52.19 | 49.56 | 76.48 | 71.86 | 72.28 | 68.15 | 60.52 | 56.26 | 55.46 | 51.95 |
| ZS CLIP | 50.08 | 44.74 | 43.75 | 38.91 | 60.37 | 51.30 | 56.08 | 46.91 | 44.35 | 38.40 | 39.80 | 33.78 |
| VPT-D | 53.82 | 49.27 | 47.11 | 43.17 | 69.30 | 63.65 | 64.55 | 58.95 | 53.84 | 49.03 | 48.17 | 43.73 |
| AdaptFormer | 55.05 | 49.40 | 49.34 | 44.23 | 65.51 | 57.30 | 61.09 | 52.89 | 49.33 | 43.38 | 44.94 | 38.67 |
| IVLP | 56.59 | 51.91 | 50.09 | 45.99 | 68.32 | 62.44 | 63.84 | 58.02 | 47.21 | 41.41 | 42.73 | 37.15 |
| IVLA | 55.66 | 50.16 | 49.92 | 44.94 | 65.73 | 56.42 | 62.34 | 54.42 | 50.22 | 44.12 | 45.50 | 39.82 |
| MaPLe | **59.56** | 56.02 | **52.85** | 49.63 | 75.67 | 70.64 | 70.83 | 65.92 | 59.09 | 54.77 | 53.32 | 49.35 |
| MMA | 56.62 | 51.37 | 50.72 | 45.98 | 67.95 | 60.25 | 63.47 | 55.82 | 51.94 | 46.19 | 46.73 | 41.31 |
| **MAIL** | 59.48 | **56.29** | 52.83 | **49.99** | **77.19** | **72.13** | **72.53** | **67.72** | **60.40** | **56.12** | **54.70** | **50.87** |

Table 18: 1-shot $U^D$CDR evaluation results (%) on DomainNet.

| Methods | Sketch | | Quickdraw | | Painting | | Infograph | | Clipart | | Average | |
| | $mAP_{200}$ | $Prec_{200}$ | $mAP_{200}$ | $Prec_{200}$ | $mAP_{200}$ | $Prec_{200}$ | $mAP_{200}$ | $Prec_{200}$ | $mAP_{200}$ | $Prec_{200}$ | $mAP_{200}$ | $Prec_{200}$ |
|---|---|---|---|---|---|---|---|---|---|---|---|---|
| ProS* | 73.85 | 49.11 | 28.89 | 11.86 | 72.27 | 46.15 | 60.56 | 39.62 | 81.05 | 52.98 | 63.32 | 39.94 |
| ZS CLIP | 47.60 | 28.71 | 8.67 | 4.50 | 55.69 | 31.70 | 47.56 | 29.36 | 55.81 | 31.10 | 43.07 | 25.07 |
| VPT-D | 62.69 | 40.36 | 20.04 | 8.91 | 63.44 | 38.43 | 55.64 | 35.40 | 69.49 | 42.50 | 54.26 | 33.12 |
| AdaptFormer | 55.36 | 33.66 | 11.75 | 5.63 | 58.74 | 32.41 | 54.15 | 31.65 | 61.41 | 34.99 | 48.28 | 27.67 |
| IVLP | 59.60 | 38.37 | 15.97 | 7.73 | 63.40 | 37.86 | 57.27 | 35.49 | 67.76 | 40.84 | 52.80 | 32.05 |
| IVLA | 57.31 | 34.97 | 12.81 | 5.99 | 59.59 | 33.05 | 55.07 | 32.35 | 62.83 | 35.98 | 49.52 | 28.47 |
| MaPLe | 68.54 | 45.29 | 25.18 | 10.80 | 69.06 | 43.30 | 60.06 | 39.10 | 76.15 | 48.14 | 59.79 | 37.33 |
| MMA | 59.39 | 36.31 | 14.38 | 6.40 | 60.67 | 33.98 | 56.20 | 33.27 | 64.27 | 37.02 | 50.98 | 29.39 |
| **MAIL** | **70.28** | **45.79** | **25.79** | **11.06** | **70.15** | **43.15** | **60.19** | **39.09** | **77.10** | **48.66** | **60.70** | **37.61** |

Table 19: 1-shot $U^C$CDR evaluation results (%) on Sketchy and TU-Berlin.

| Methods | Sketchy | | TU-Berlin | |
| | $mAP_{200}$ | $Prec_{200}$ | $mAP_{all}$ | $Prec_{100}$ |
|---|---|---|---|---|
| ProS* | 69.91 | 65.45 | 66.75 | 74.42 |
| ZS CLIP | 35.82 | 33.08 | 31.45 | 46.12 |
| VPT-D | 58.43 | 53.74 | 57.91 | 67.81 |
| AdaptFormer | 52.15 | 48.87 | 46.38 | 60.52 |
| IVLP | 51.98 | 47.76 | 53.33 | 65.16 |
| IVLA | 52.17 | 48.89 | 46.85 | 60.76 |
| MaPLe | **65.95** | **61.51** | 63.00 | 71.85 |
| MMA | 52.53 | 49.31 | 50.26 | 63.68 |
| **MAIL** | 65.70 | 61.24 | **65.30** | **73.82** |

## I.2  3-Shot Results

Table 20: 3-shot UCDR evaluation results (%) on DomainNet.

| Methods | Sketch | | | | Quickdraw | | | | Painting | | | |
| | Unseen Gallery | | MixedGallery | | UnseenGallery | | MixedGallery | | UnseenGallery | | MixedGallery | |
| | $mAP_{200}$ | $Prec_{200}$ | $mAP_{200}$ | $Prec_{200}$ | $mAP_{200}$ | $Prec_{200}$ | $mAP_{200}$ | $Prec_{200}$ | $mAP_{200}$ | $Prec_{200}$ | $mAP_{200}$ | $Prec_{200}$ |
|---|---|---|---|---|---|---|---|---|---|---|---|---|
| ProS* | 64.67 | 60.01 | 58.43 | 54.63 | 28.42 | 25.44 | 23.18 | 21.27 | 75.16 | 69.55 | 71.20 | 66.12 |
| ZS CLIP | 42.20 | 35.28 | 36.62 | 29.79 | 7.44 | 5.61 | 6.00 | 3.17 | 61.68 | 55.07 | 56.53 | 50.14 |
| VPT-D | 60.47 | 55.78 | 53.00 | 48.80 | 23.83 | 21.17 | 18.16 | 16.44 | 73.32 | 67.32 | 68.20 | 62.61 |
| AdaptFormer | 64.44 | 59.53 | 57.38 | 53.05 | 27.28 | 24.45 | 21.75 | 19.70 | 74.47 | 68.63 | 69.67 | 64.27 |
| IVLP | 59.66 | 55.05 | 52.37 | 48.30 | 22.53 | 20.13 | 16.77 | 15.27 | 72.64 | 66.89 | 67.50 | 62.16 |
| IVLA | 63.77 | 59.91 | 57.68 | 53.40 | 26.16 | 24.31 | 20.58 | 19.57 | 74.30 | 68.52 | 69.44 | 64.10 |
| MaPLe | 64.38 | 59.88 | 57.46 | 53.48 | 26.64 | 23.86 | 21.06 | 19.14 | 75.55 | 70.00 | 70.91 | 65.79 |
| MMA | 65.99 | 61.55 | 59.04 | 55.11 | 27.61 | 25.19 | 21.79 | 20.29 | 75.52 | 70.07 | 70.70 | 65.68 |
| **MAIL** | **67.68** | **63.63** | **60.81** | **57.24** | **28.15** | **25.70** | **22.24** | **20.55** | **77.22** | **72.07** | **72.35** | **67.69** |

| Methods | Infograph | | | | Clipart | | | | Average | | | |
| | UnseenGallery | | MixedGallery | | UnseenGallery | | MixedGallery | | UnseenGallery | | MixedGallery | |
| | $mAP_{200}$ | $Prec_{200}$ | $mAP_{200}$ | $Prec_{200}$ | $mAP_{200}$ | $Prec_{200}$ | $mAP_{200}$ | $Prec_{200}$ | $mAP_{200}$ | $Prec_{200}$ | $mAP_{200}$ | $Prec_{200}$ |
|---|---|---|---|---|---|---|---|---|---|---|---|---|
| ProS* | 57.98 | 54.42 | 52.19 | 49.56 | 76.48 | 71.86 | 72.28 | 68.15 | 60.52 | 56.26 | 55.46 | 51.95 |
| ZS CLIP | 50.08 | 44.74 | 43.75 | 38.91 | 60.37 | 51.30 | 56.08 | 46.91 | 44.35 | 38.40 | 39.80 | 33.78 |
| VPT-D | 57.70 | 53.82 | 50.72 | 47.29 | 73.94 | 68.99 | 68.81 | 63.98 | 57.85 | 53.42 | 51.78 | 47.82 |
| AdaptFormer | 57.55 | 53.59 | 51.13 | 47.47 | 77.24 | 72.10 | 72.44 | 67.56 | 60.19 | 55.66 | 54.47 | 50.41 |
| IVLP | 58.95 | 55.10 | 52.11 | 48.68 | 73.44 | 68.38 | 68.57 | 63.63 | 57.44 | 53.11 | 51.46 | 47.61 |
| IVLA | 56.62 | 52.61 | 50.08 | 47.39 | 76.27 | 72.02 | 72.44 | 67.48 | 59.42 | 55.47 | 54.04 | 50.39 |
| MaPLe | 60.65 | 57.36 | 53.94 | 51.08 | 77.08 | 72.30 | 71.99 | 67.56 | 60.86 | 56.68 | 55.07 | 51.41 |
| MMA | 58.08 | 56.40 | 51.46 | 48.16 | 77.92 | 73.43 | 72.97 | 68.81 | 61.02 | 56.93 | 55.19 | 51.61 |
| **MAIL** | **61.65** | **58.81** | **55.06** | **52.70** | **79.29** | **75.31** | **74.61** | **71.05** | **62.80** | **59.10** | **57.01** | **53.85** |

Table 21: 3-shot $U^{D}$CDR evaluation results (%) on DomainNet.

| Methods | Sketch | | Quickdraw | | Painting | | Infograph | | Clipart | | Average | |
| | $mAP_{200}$ | $Prec_{200}$ | $mAP_{200}$ | $Prec_{200}$ | $mAP_{200}$ | $Prec_{200}$ | $mAP_{200}$ | $Prec_{200}$ | $mAP_{200}$ | $Prec_{200}$ | $mAP_{200}$ | $Prec_{200}$ |
|---|---|---|---|---|---|---|---|---|---|---|---|---|
| ProS* | 73.85 | 49.11 | 28.89 | 11.86 | 72.27 | 46.15 | 60.56 | 39.62 | 81.05 | 52.98 | 63.32 | 39.94 |
| ZS CLIP | 47.60 | 28.71 | 8.67 | 4.50 | 55.69 | 31.70 | 47.56 | 29.36 | 55.81 | 31.10 | 43.07 | 25.07 |
| VPT-D | 68.21 | 45.24 | 22.76 | 10.09 | 68.17 | 42.51 | 60.59 | 39.10 | 75.14 | 47.89 | 58.61 | 36.75 |
| AdaptFormer | 71.93 | 46.63 | 24.77 | 10.68 | 69.15 | 41.88 | 59.36 | 38.39 | 76.87 | 47.61 | 60.42 | 37.03 |
| IVLP | 67.46 | 44.41 | 22.68 | 9.66 | 67.96 | 42.08 | 60.15 | 38.30 | 74.37 | 46.90 | 58.52 | 36.27 |
| IVLA | 72.16 | 46.62 | 24.94 | 10.64 | 68.88 | 41.69 | 59.19 | 38.13 | 76.80 | 47.41 | 60.39 | 36.90 |
| MaPLe | 71.91 | 47.27 | 25.92 | 10.82 | 71.41 | 44.92 | 61.97 | 39.85 | 78.29 | 49.38 | 61.90 | 38.45 |
| MMA | 73.44 | 48.39 | 26.53 | 11.27 | 70.84 | 43.94 | 60.43 | 39.38 | 78.86 | 50.33 | 62.02 | 38.66 |
| **MAIL** | **75.65** | **50.54** | **27.30** | **11.24** | **74.18** | **47.23** | **64.37** | **41.91** | **80.99** | **53.45** | **64.50** | **40.87** |

Table 22: 3-shot $U^{C}$CDR evaluation results (%) on Sketchy and TU-Berlin.

| Methods | Sketchy | | TU-Berlin | |
| | $mAP_{200}$ | $Prec_{200}$ | $mAP_{all}$ | $Prec_{100}$ |
|---|---|---|---|---|
| ProS* | 69.91 | 65.45 | 66.75 | 74.42 |
| ZS CLIP | 35.82 | 33.08 | 31.45 | 46.12 |
| VPT-D | 67.03 | 63.32 | 63.35 | 71.52 |
| AdaptFormer | 63.97 | 58.79 | 65.44 | 73.22 |
| IVLP | 64.72 | 60.48 | 60.95 | 69.54 |
| IVLA | 64.36 | 59.12 | 64.97 | 73.02 |
| MaPLe | 73.28 | 69.44 | 66.72 | 73.93 |
| MMA | 69.74 | 65.14 | 67.39 | 74.12 |
| **MAIL** | **75.73** | **71.61** | **68.28** | **74.51** |

## I.3  4-Shot Results

Table 23: 4-shot UCDR evaluation results (%) on DomainNet.

| Methods | Sketch | | | | Quickdraw | | | | Painting | | | |
|---|---|---|---|---|---|---|---|---|---|---|---|---|
| | Unseen Gallery | | MixedGallery | | UnseenGallery | | MixedGallery | | UnseenGallery | | MixedGallery | |
| | mAP$_{200}$ | Prec$_{200}$ | mAP$_{200}$ | Prec$_{200}$ | mAP$_{200}$ | Prec$_{200}$ | mAP$_{200}$ | Prec$_{200}$ | mAP$_{200}$ | Prec$_{200}$ | mAP$_{200}$ | Prec$_{200}$ |
| ProS* | 64.67 | 60.01 | 58.43 | 54.63 | 28.42 | 25.44 | 23.18 | 21.27 | 75.16 | 69.55 | 71.20 | 66.12 |
| ZS CLIP | 42.20 | 35.28 | 36.62 | 29.79 | 7.44 | 5.61 | 6.00 | 3.17 | 61.68 | 55.07 | 56.53 | 50.14 |
| VPT-D | 62.30 | 57.65 | 54.61 | 50.52 | 26.01 | 23.32 | 19.96 | 18.27 | 74.32 | 68.36 | 69.21 | 63.70 |
| AdaptFormer | 65.22 | 61.11 | 58.58 | 54.73 | 28.83 | 26.07 | 22.36 | 20.89 | 75.11 | 69.52 | 70.33 | 65.18 |
| IVLP | 62.06 | 57.59 | 54.58 | 50.72 | 23.83 | 21.38 | 18.19 | 16.68 | 73.64 | 67.91 | 68.47 | 63.15 |
| IVLA | 65.57 | 61.04 | 58.52 | 54.57 | 27.24 | 25.83 | 21.83 | 20.81 | 74.97 | 69.40 | 70.09 | 64.95 |
| MaPLe | 65.76 | 61.43 | 58.61 | 54.86 | 28.46 | 25.80 | 22.42 | 20.72 | 75.41 | 69.86 | 70.76 | 65.62 |
| MMA | 66.65 | 62.36 | 59.80 | 55.99 | 28.54 | 26.20 | 22.70 | 21.22 | 75.49 | 70.04 | 70.64 | 65.66 |
| **MAIL** | **68.74** | **64.64** | **61.93** | **58.39** | **30.24** | **27.82** | **23.77** | **22.37** | **77.37** | **72.20** | **72.46** | **67.80** |

| Methods | Infograph | | | | Clipart | | | | Average | | | |
|---|---|---|---|---|---|---|---|---|---|---|---|---|
| | UnseenGallery | | MixedGallery | | UnseenGallery | | MixedGallery | | UnseenGallery | | MixedGallery | |
| | mAP$_{200}$ | Prec$_{200}$ | mAP$_{200}$ | Prec$_{200}$ | mAP$_{200}$ | Prec$_{200}$ | mAP$_{200}$ | Prec$_{200}$ | mAP$_{200}$ | Prec$_{200}$ | mAP$_{200}$ | Prec$_{200}$ |
| ProS* | 57.98 | 54.42 | 52.19 | 49.56 | 76.48 | 71.86 | 72.28 | 68.15 | 60.52 | 56.26 | 55.46 | 51.95 |
| ZS CLIP | 50.08 | 44.74 | 43.75 | 38.91 | 60.37 | 51.30 | 56.08 | 46.91 | 44.35 | 38.40 | 39.80 | 33.78 |
| VPT-D | 59.10 | 55.35 | 52.14 | 48.79 | 75.28 | 70.46 | 70.20 | 65.55 | 59.40 | 55.03 | 53.22 | 49.37 |
| AdaptFormer | 57.14 | 53.44 | 50.65 | 47.31 | 78.11 | 73.60 | 73.21 | 69.05 | 60.94 | 56.74 | 55.02 | 51.43 |
| IVLP | 59.99 | 56.25 | 53.21 | 49.86 | 74.77 | 69.82 | 69.88 | 65.12 | 58.86 | 54.59 | 52.87 | 49.10 |
| IVLA | 56.77 | 54.03 | 51.20 | 47.78 | 77.08 | 72.35 | 72.19 | 68.79 | 60.33 | 56.53 | 54.76 | 51.38 |
| MaPLe | 60.48 | 57.19 | 53.79 | 50.80 | 77.15 | 72.74 | 72.02 | 67.92 | 61.45 | 57.40 | 55.52 | 51.98 |
| MMA | 59.93 | 56.59 | 53.50 | 50.53 | 78.27 | 73.90 | 73.35 | 69.34 | 61.78 | 57.82 | 55.60 | 52.55 |
| **MAIL** | **62.34** | **59.59** | **55.67** | **53.37** | **79.50** | **75.46** | **74.75** | **71.22** | **63.64** | **59.94** | **57.71** | **54.63** |

Table 24: 4-shot U$^D$CDR evaluation results (%) on DomainNet.

| Methods | Sketch | | Quickdraw | | Painting | | Infograph | | Clipart | | Average | |
|---|---|---|---|---|---|---|---|---|---|---|---|---|
| | mAP$_{200}$ | Prec$_{200}$ | mAP$_{200}$ | Prec$_{200}$ | mAP$_{200}$ | Prec$_{200}$ | mAP$_{200}$ | Prec$_{200}$ | mAP$_{200}$ | Prec$_{200}$ | mAP$_{200}$ | Prec$_{200}$ |
| ProS* | 73.85 | 49.11 | 28.89 | 11.86 | 72.27 | 46.15 | 60.56 | 39.62 | 81.05 | 52.98 | 63.32 | 39.94 |
| ZS CLIP | 47.60 | 28.71 | 8.67 | 4.50 | 55.69 | 31.70 | 47.56 | 29.36 | 55.81 | 31.10 | 43.07 | 25.07 |
| VPT-D | 70.43 | 46.26 | 25.21 | 10.91 | 69.31 | 43.01 | 60.98 | 39.39 | 76.52 | 48.51 | 60.49 | 37.62 |
| AdaptFormer | 73.37 | 48.15 | 27.03 | 11.60 | 70.91 | 43.90 | 59.57 | 39.05 | 79.06 | 50.38 | 61.99 | 38.62 |
| IVLP | 69.90 | 45.58 | 24.20 | 10.29 | 68.80 | 42.40 | 61.58 | 39.34 | 75.19 | 47.25 | 59.93 | 36.97 |
| IVLA | 73.25 | 47.76 | 26.77 | 11.22 | 70.37 | 43.28 | 59.78 | 38.88 | 78.66 | 49.65 | 61.76 | 38.19 |
| MaPLe | 73.59 | 48.62 | 28.01 | 11.54 | 71.59 | 45.26 | 61.88 | 40.59 | 79.32 | 51.00 | 62.88 | 39.40 |
| MMA | 74.39 | 48.85 | 27.14 | 11.42 | 71.40 | 44.54 | 61.18 | 39.86 | 79.31 | 50.86 | 62.68 | 39.11 |
| **MAIL** | **76.26** | **50.56** | **29.31** | **11.95** | **74.98** | **47.53** | **65.33** | **42.44** | **82.03** | **53.52** | **65.58** | **41.20** |

Table 25: 4-shot U$^C$CDR evaluation results (%) on Sketchy and TU-Berlin.

| Methods | Sketchy | | TU-Berlin | |
|---|---|---|---|---|
| | mAP$_{200}$ | Prec$_{200}$ | mAP$_{all}$ | Prec$_{100}$ |
| ProS* | 69.91 | 65.45 | 66.75 | 74.42 |
| ZS CLIP | 35.82 | 33.08 | 31.45 | 46.12 |
| VPT-D | 71.55 | 67.32 | 67.16 | 74.78 |
| AdaptFormer | 69.76 | 65.27 | 67.64 | 73.98 |
| IVLP | 66.36 | 62.39 | 62.83 | 70.88 |
| IVLA | 69.79 | 65.18 | 66.89 | 73.94 |
| MaPLe | 73.54 | 69.60 | 66.76 | 73.84 |
| MMA | 72.20 | 68.17 | 68.22 | 74.46 |
| **MAIL** | **75.51** | **71.91** | **68.77** | **74.94** |

## I.4  8-Shot Results

Table 26: 8-shot UCDR evaluation results (%) on DomainNet.

| Methods | Sketch | | | | Quickdraw | | | | Painting | | | |
| | Unseen Gallery | | MixedGallery | | UnseenGallery | | MixedGallery | | UnseenGallery | | MixedGallery | |
| | mAP$_{200}$ | Prec$_{200}$ | mAP$_{200}$ | Prec$_{200}$ | mAP$_{200}$ | Prec$_{200}$ | mAP$_{200}$ | Prec$_{200}$ | mAP$_{200}$ | Prec$_{200}$ | mAP$_{200}$ | Prec$_{200}$ |
|---|---|---|---|---|---|---|---|---|---|---|---|---|
| ProS* | 64.67 | 60.01 | 58.43 | 54.63 | 28.42 | 25.44 | 23.18 | 21.27 | 75.16 | 69.55 | 71.20 | 66.12 |
| ZS CLIP | 42.20 | 35.28 | 36.62 | 29.79 | 7.44 | 5.61 | 6.00 | 3.17 | 61.68 | 55.07 | 56.53 | 50.14 |
| VPT-D | 64.28 | 59.77 | 56.67 | 52.80 | 28.67 | 26.08 | 22.06 | 20.20 | 75.48 | 69.76 | 70.35 | 64.72 |
| AdaptFormer | 65.95 | 62.09 | 59.03 | 55.61 | 28.11 | 26.20 | 21.80 | 20.52 | 75.15 | 69.91 | 70.55 | 65.70 |
| IVLP | 64.37 | 59.90 | 56.90 | 53.07 | 27.66 | 25.13 | 21.15 | 19.63 | 75.10 | 69.54 | 69.93 | 64.82 |
| IVLA | 67.13 | 62.89 | 60.22 | 56.53 | 28.78 | 26.59 | 22.72 | 21.36 | 75.85 | 70.43 | 71.09 | 66.14 |
| MaPLe | 66.37 | 62.25 | 59.26 | 55.77 | 29.83 | 27.29 | 23.35 | 21.75 | 75.16 | 69.48 | 70.37 | 65.19 |
| MMA | 67.44 | 63.24 | 60.67 | 57.04 | 29.58 | 27.29 | 23.40 | 21.94 | 75.78 | 70.43 | 71.05 | 66.16 |
| **MAIL** | **68.72** | **65.01** | **62.22** | **58.56** | **31.41** | **29.06** | **24.08** | **22.79** | **77.21** | **72.15** | **72.41** | **67.62** |

| Methods | Infograph | | | | Clipart | | | | Average | | | |
| | UnseenGallery | | MixedGallery | | UnseenGallery | | MixedGallery | | UnseenGallery | | MixedGallery | |
| | mAP$_{200}$ | Prec$_{200}$ | mAP$_{200}$ | Prec$_{200}$ | mAP$_{200}$ | Prec$_{200}$ | mAP$_{200}$ | Prec$_{200}$ | mAP$_{200}$ | Prec$_{200}$ | mAP$_{200}$ | Prec$_{200}$ |
|---|---|---|---|---|---|---|---|---|---|---|---|---|
| ProS* | 57.98 | 54.42 | 52.19 | 49.56 | 76.48 | 71.86 | 72.28 | 68.15 | 60.52 | 56.26 | 55.46 | 51.95 |
| ZS CLIP | 50.08 | 44.74 | 43.75 | 38.91 | 60.37 | 51.30 | 56.08 | 46.91 | 44.35 | 38.40 | 39.80 | 33.78 |
| VPT-D | 60.44 | 57.05 | 53.35 | 50.36 | 77.08 | 72.45 | 71.87 | 67.50 | 61.19 | 57.02 | 54.86 | 51.12 |
| AdaptFormer | 55.72 | 52.73 | 49.17 | 46.50 | 77.87 | 73.83 | 72.70 | 69.04 | 60.56 | 56.95 | 54.65 | 51.47 |
| IVLP | 61.20 | 57.83 | 54.17 | 51.22 | 76.79 | 72.30 | 71.85 | 67.41 | 61.02 | 56.94 | 54.80 | 51.23 |
| IVLA | 58.11 | 54.87 | 51.48 | 48.55 | 78.62 | 74.32 | 73.65 | 69.73 | 61.70 | 57.82 | 55.83 | 52.46 |
| MaPLe | 60.58 | 57.46 | 53.93 | 51.04 | 77.61 | 73.03 | 72.83 | 68.60 | 61.91 | 57.90 | 55.95 | 52.47 |
| MMA | 60.29 | 57.38 | 53.83 | 51.44 | 78.48 | 74.25 | 73.49 | 69.68 | 62.31 | 58.54 | 56.49 | 53.25 |
| **MAIL** | **62.22** | **59.44** | **55.10** | **53.10** | **79.22** | **75.31** | **74.78** | **71.01** | **63.75** | **60.19** | **57.72** | **54.62** |

Table 27: 8-shot U$^D$CDR evaluation results (%) on DomainNet.

| Methods | Sketch | | Quickdraw | | Painting | | Infograph | | Clipart | | Average | |
| | mAP$_{200}$ | Prec$_{200}$ | mAP$_{200}$ | Prec$_{200}$ | mAP$_{200}$ | Prec$_{200}$ | mAP$_{200}$ | Prec$_{200}$ | mAP$_{200}$ | Prec$_{200}$ | mAP$_{200}$ | Prec$_{200}$ |
|---|---|---|---|---|---|---|---|---|---|---|---|---|
| ProS* | 73.85 | 49.11 | 28.89 | 11.86 | 72.27 | 46.15 | 60.56 | 39.62 | 81.05 | 52.98 | 63.32 | 39.94 |
| ZS CLIP | 47.60 | 28.71 | 8.67 | 4.50 | 55.69 | 31.70 | 47.56 | 29.36 | 55.81 | 31.10 | 43.07 | 25.07 |
| VPT-D | 72.76 | 48.21 | 27.66 | 10.13 | 71.22 | 44.72 | 63.77 | 41.24 | 79.12 | 50.57 | 62.91 | 38.97 |
| AdaptFormer | 75.19 | 49.58 | 28.01 | 11.79 | 71.71 | 44.90 | 60.62 | 40.12 | 80.44 | 52.42 | 63.19 | 39.76 |
| IVLP | 72.29 | 47.71 | 27.10 | 11.24 | 71.10 | 44.64 | 63.79 | 40.89 | 78.44 | 50.11 | 62.54 | 38.92 |
| IVLA | 74.90 | 49.47 | 28.05 | 11.74 | 71.80 | 44.88 | 60.99 | 40.04 | 80.23 | 51.53 | 63.19 | 39.53 |
| MaPLe | 74.60 | 49.62 | 29.54 | 12.08 | 71.84 | 45.27 | 63.06 | 40.93 | 79.45 | 51.23 | 63.70 | 39.75 |
| MMA | 75.38 | 49.97 | 28.52 | 11.87 | 72.26 | 45.12 | 62.82 | 41.14 | 80.38 | 51.66 | 63.87 | 39.95 |
| **MAIL** | **76.87** | **51.53** | **30.76** | **12.67** | **75.47** | **48.22** | **65.79** | **42.92** | **81.99** | **53.74** | **66.17** | **41.81** |

Table 28: 8-shot U$^C$CDR evaluation results (%) on Sketchy and TU-Berlin.

| Methods | Sketchy | | TU-Berlin | |
| | mAP$_{200}$ | Prec$_{200}$ | mAP$_{all}$ | Prec$_{100}$ |
|---|---|---|---|---|
| ProS* | 69.91 | 65.45 | 66.75 | 74.42 |
| ZS CLIP | 35.82 | 33.08 | 31.45 | 46.12 |
| VPT-D | 70.86 | 67.19 | 66.79 | 74.44 |
| AdaptFormer | 74.51 | 70.98 | 68.42 | 73.24 |
| IVLP | 70.08 | 66.24 | 65.23 | 72.83 |
| IVLA | 74.48 | 70.70 | 68.84 | 74.46 |
| MaPLe | 74.86 | 70.98 | 67.66 | 74.27 |
| MMA | 75.08 | 71.63 | **68.94** | **74.66** |
| **MAIL** | **76.22** | **72.78** | 68.91 | 74.61 |

