# OpenReview forum: "Multi-Modal Interactive Agent Layer for Few-Shot Universal Cross-Domain Retrieval and Beyond"
_NeurIPS.cc/2025/Conference — NeurIPS 2025 poster_

### Official Review · Reviewer_8mcV · 2025-06-25

**Clarity:** 2
**Significance:** 3
**Originality:** 3
**Rating:** 4
**Confidence:** 3

**Summary:**

This paper focuses on few-shot universal cross-domain retrieval (FS-UCDR), a newly introduced scenario that considers limited data in traditional universal cross-domain retrieval. Authors experimentally find that modality-coupled methods work well under the proposed FS-UCDR to parameter-efficiently adapt VLMs for this task, and correspondingly design the Multi-Modal Interactive Agent Layer (MAIL) to align parameter updates of different layers. Extensive experiments on retrieval and few-shot classification validate the effectiveness and efficiency of the proposed method.

**Questions:**

1. Given the non-trivial nature of the partially fine-tuned methods mentioned in the paper, why still try to introduce modality coupling to such methods? Are there any reasons other than the under-explored attempts for including modality coupling in partially fine-tuned methods?

2. What's the connection between the MAIL and FS-UCDR? Why would it address the difficulties of generalizing with limited training data?

3. What's the rationale behind aligning parameter updates? Why would it help the adaptation of VLMs? More analyses are encouraged to be provided.

Overall, my concerns focus on the motivation of the designed approach, the connection with FS-UCDR, and the rationale behind aligning parameter updates. I would appreciate the authors’ clarifications on these questions. Provided that the requested clarifications are addressed, I will improve the recommendation score.

**Ethical Concerns:**

["NO or VERY MINOR ethics concerns only"]

**Final Justification:**

The rebuttal has addressed my concerns over the motivation of the designed approach, the connection with FS-UCDR, and the rationale behind aligning parameter updates. Therefore, I'm raising my score to 4.

**Limitations:**

Yes

**Paper Formatting Concerns:**

No formatting issues.

**Quality:**

2

**Strengths And Weaknesses:**

Strength:

1. The overall structure is clear and easy to follow.

2. The newly introduced FS-UCDR scenario is practical, exploring whether a few samples from each source domain can lead to good generalization performance.

3. Experiments are solid. MAIL is evaluated both on FS-UCDR and few-shot classification, with results on multiple datasets supporting the effectiveness.

Weakness:

1. The motivation for introducing modality-coupling into the partially fine-tuned methods is unclear. Authors claim that the lack of such endeavors in the literature derives from the non-trivial nature. However, a hard-to-implement design does not necessarily guarantee better performance. Therefore, more clarifications are required.

2. It seems that the authors introduce a general PEFT method that works for all tasks that require the adaptation of VLMs. Motivation mainly focuses on inference efficiency and the fact that modality-coupling has not been explored in the fine-tuned-based approaches. Little motivation is given on why this is a good design for Universal Cross-Domain Retrieval and the newly introduced Few-Shot-UCDR.

3. It's quite confusing for readers to understand the original layer, target layer, and agent layer without context and specifications. Please define earlier in the Introduction Section.

---

> ### Author Rebuttal · Authors · 2025-07-30
>
> **R-1: Weakness1 & Question1**
>
> We appreciate the reviewer's question. Our motivations for introducing modality-coupling are as follows:
>
> > 1.  The primary goal of vision-language models is to align image and text modalities within a shared feature space. In CLIP, this alignment is achieved through dual encoders jointly trained on 400M image-text pairs during pre-training.
> >2. When adapting to downstream tasks （typically the few-shot classification problem） or new domains, maintaining this cross-modal alignment remains crucial. Early approaches like CoOp [1] fine-tune only the text encoder,  which may disrupt the alignment between the image and text encoders under the few-shot setting. Therefore, MaPLe [2] and MMA [3] propose to  fine-tune both the text and image encoders such  that their optimal alignment can be achieved on the downstream domains.
> > 3. **Role of Modality Coupling:**  As demonstrated in MaPLe and MMA, explicitly modeling the interaction between image and text branches (modality coupling) during fine-tuning improves representational coherence. Such coupling ensures that updates to one modality are reflected in the other, thereby promoting more consistent cross-modal alignment.
> > 4. As demonstrated in both the MaPLe and MMA papers, as well as in our own work, modality coupling proves effective in enhancing performance for both few-shot classification and FS-UCDR. It is therefore natural for us to explore how modality coupling can be incorporated into the third category of PEFT methods—namely, partially fine-tuned methods. This motivation led to the development of MAIL, which achieves state-of-the-art performance on both few-shot classification and FS-UCDR tasks.
>
> In summary, we argue that any fine-tuning method intended to adapt a foundational multi-modal model like CLIP should preserve the alignment established during pre-training. Introducing modality coupling is a principled extension of this objective,  supported by empirical evidence from works like MaPLe and MMA. If the reviewer still has concerns about the motivation, we will try to explain from a different perspective (the regulation effect) in **R-3**.
>
>
>
> **R-2: Weakness2 & Question2**
>
> We apologize for any misunderstanding caused by our writing. We would like to offer the following clarifications:
>
> > 1. CLIP-based PEFT methods have demonstrated remarkable success in few-shot classification scenarios with **limited training data**, primarily due to CLIP's strong generalization capabilities.  In fact, the challenge of generalizing under limited supervision is largely mitigated by the powerful CLIP backbone, while subsequent PEFT methods—including our proposed MAIL—serve to further enhance CLIP’s adaptability and performance.
> > 2. Current CLIP-based PEFT methods mainly test on the few-shot classification scenario. We identify the *Universal Cross-Domain Retrieval* (UCDR) setting, which provides a more challenging and informative benchmark for evaluating the generalization ability of PEFT frameworks. Specifically, UCDR requires models to generalize to both unseen domains and unseen classes during testing, making it a more rigorous standard for assessing cross-domain and cross-category generalization. This observation motivated us to explore the potential of CLIP-based PEFT methods in addressing the FS-UCDR setting.
> > 3. We **empirically** demonstrate that modality-coupled methods are good designs for FS-UCDR, and MAIL is a general-purpose modality-coupled PEFT framework proposed in our work,  it is not designed exclusively for FS-UCDR. Rather, our goal is to provide a flexible framework that can be adapted to a range of few-shot tasks.
> > 4. Conceptually, we could have written a standalone paper focused solely on few-shot classification with MAIL, similar to works like MaPLe and MMA. However, we chose to broaden the scope to offer a more comprehensive study, which we believe not only benefits the PEFT community but also benefits the UCDR community—particularly under low-resource conditions.
> > 5. Moreover, one can design MAIL variants specifically tailored to FS-UCDR. For example, introducing domain-specific agent layers may be beneficial, given that UCDR inherently involves multiple domains.
>
>
>
> **R-3:  Question3**
>
> - If the reviewer is expecting a theoretical or mathematical framework, we sincerely apologize that we are currently unable to provide one. However, as this work is submitted to the Vision-Language Application track, we believe that thorough experimental validation and strong empirical results should be given greater emphasis in this context.
>
>   > 1. In fact, methods such as MaPLe [2] and MMA [3] also pursue the alignment of parameter updates. The key difference lies in the scope of alignment: they focus on aligning the updates of newly introduced parameters (e.g., prompts or adapters), whereas our MAIL aims to align updates of internal parameters (i.e., target layers) within the pre-trained backbone.
>   >
>   > 2. It is worth noting that the original MaPLe and MMA papers do not provide a formal theoretical justification for why aligning parameter updates improves performance. Instead, they argue that such alignment leads to more **sufficient** and **complete** adaptation of vision-language models—an intuition rooted in the inherently multi-modal nature of VLMs. Our work builds on this intuition by extending it to a different fine-tuning regime.
>
> - Here we would like to the rationale behind aligning parameter updates in an intuitive way.
>
>   > When fine-tuning CLIP, a key challenge is to prevent overfitting. As shown in our Equation (11), one of the most common and straightforward approaches is to apply an regulation loss separately on the outputs of the two modality-specific heads:
>   > $$
>   > L_{reg}^v=f(V, V^F), L_{reg}^t=f(T, T^F)
>   > $$
>   > where $V$ and $T$ denote the visual and textual outputs of the frozen CLIP, while $V^F$ and $T^F$ denote the fine-tuned outputs given the same input. $f(\cdot)$ denotes the regulation loss, typically the $L_2$ loss.
>   >
>   > - We argue that  aligning parameter updates can be also viewed as a form of regularization. Such coupling ensures that updates to one modality are reflected in the other, thereby encouraging more **consistent** cross-modal alignment.
>   >
>   > - This explains why modality-coupled methods perform well in the FS-UCDR setting. In FS-UCDR, both the domains and classes encountered during testing are unseen during training, which requires the model to possess strong generalization capabilities.
>   >
>   > - This also helps to interpret the results in Table 4 from our main paper. While our method achieves comparable accuracy to DeKg (ICLR’25) on the **base** classes, it yields a **significant** improvement on the **new** classes. The “new” setting refers to test classes that are not present during training, thereby placing higher demands on the model’s generalization ability—precisely where our approach excels.
>
>
>
> **R-4: Weakness 3**
>
> We sincerely thank the reviewer for pointing out this issue. We will address it carefully in the revised version. We also apologize for any confusion that may have arisen during your review.
>
> Once again, we greatly appreciate the reviewer's insightful comments. We hope our responses could address your concerns. If you have any additional questions or suggestions, please let us know.
>
>
>
> [1] Kaiyang Zhou, Jingkang Yang, et al. Learning to prompt for vision-language models. IJCV, 2022.
>
> [2] Muhammad Uzair Khattak, Hanoona Rasheed, et al. Maple: Multi-modal prompt learning. CVPR, 2023.
>
> [3] Lingxiao Yang, Ru-Yuan Zhang, et al. Mma: Multi-modal adapter for  vision-language models. CVPR, 2024.

---

> > ### Comment · Reviewer_8mcV · 2025-08-04
> > **Response to rebuttal**
> >
> > Thank you for the response. I now better understand the role of Modality Coupling and the potential of extending this to the third category of PEFT methods. I also appreciate the authors’ clarification regarding the rationale for aligning parameter updates from a regularization perspective.
> >
> > Given these explanations, I will raise my score to 4. I look forward to seeing the corresponding updates in the revised version of the paper.

---

> > > ### Author Response · Authors · 2025-08-05
> > >
> > > We thank the reviewer 8mcV for the time and effort in reviewing our rebuttal and we sincerely appreciate the reviewer for thorough and detailed review of our paper.
> > >
> > > We are delighted see to our response addresses the concerns of the reviewer and allows the reviewer to raise the rating about the assessment of the paper! We are more than happy to answer any concerns or questions the reviewer might still hold during the discussion period. Please do not hesitate to let us know!

---

### Official Review · Reviewer_ZXLZ · 2025-06-30

**Clarity:** 2
**Significance:** 2
**Originality:** 2
**Rating:** 3
**Confidence:** 3

**Summary:**

This paper proposes a new method for cross-model interaction in visual language models (in particular CLIP like models) through synchronous parameter updates for the two modalities. The main difference with existing methods is that the alignment is operating at a higher level i.e. more at the semantic level than the "signal" level. The method is particularly targeting few-shot universal cross-domain retrieval. Results on a number of datasets are showing that the method outperforms existing methods most of the time.

**Questions:**

- See the final weakness. Why is it intrinsically interesting to have this method?
- Why doesn't the reduced number of parameters not lead to more efficiency?

**Ethical Concerns:**

["NO or VERY MINOR ethics concerns only"]

**Final Justification:**

The proposed method has potential but the paper and the rebuttal do not sufficiently make clear why the method is needed instead of saying that it is a new method with similar or slightly better performance. I also feel that there are still some issues in terms of indicating performance e.g. in Table 2 Lora has better precision in many cases while the new method is being indicated in bold to have the best performance.

**Limitations:**

One limitation is indicated, so quite minimal.

**Paper Formatting Concerns:**

No issues

**Quality:**

2

**Strengths And Weaknesses:**

Strengths:
- New approach to the problem, by linking the two modalities at a different level.
- The method at inference remains to have the same complexity, no significant additional overhead.
- Results on existing benchmark compared to other methods is good.

Weaknesses:
- Performance not that different from MMA with same number of tunable parameters.
- Figure 7 very misleading. Y-axis is not starting at 0 (which for a barchart should always be the case) and y-range are not even given.
- Figure 7 indicates there is no benefit in using this method apart from the improvements in accuracy. So also figure 1 is misleading.
- It is interesting to move the alignment to higher levels with less parameters. But why would that be interesting, if training time is still higher and testing time only slightly better.

---

> ### Author Rebuttal · Authors · 2025-07-30
>
> **R-1: Weakness-1**
>
> We respectfully disagree with the claim that the performance is not significantly different from MMA under the same number of tunable parameters.
>
> > - First, as shown in **Table 1**, our method achieves consistent improvements on the 2-shot setting of DomainNet. Specifically, it achieves a **1.74% absolute gain in average mAP** and a **2.69% gain in precision** compared to MMA.  Such margins are **non-trivial in few-shot settings**, where performance improvements are generally hard to obtain due to limited supervision.
> > - More importantly, on the widely adopted and **more challenging few-shot classification benchmark**, our method outperforms MMA by a **1.23% gain in average harmonic accuracy**, as shown in **Table 4**. With the inclusion of the proposed regularization term, this gap further increases to **1.89%**, averaged over **11 diverse datasets**. This demonstrates the robustness and generalization ability of our approach across domains and tasks.
> > - Here, we provide a simpler example by listing the performance of recent methods from top-tier conferences on few-shot classification. The performance gain of MAIL over DeKg is already quite notable—especially when compared to the typical improvements between methods from 2023 to 2024. The gap becomes even more striking when compared to MMA.
> >
> > | Methods             | Base  | New   | HM    |
> > | ------------------- | ----- | ----- | ----- |
> > | CLIP [ICML'21]      | 69.34 | 74.22 | 71.70 |
> > | CoCoOp [CVPR'22]    | 80.47 | 71.69 | 75.83 |
> > | MaPLe [CVPR'23]     | 82.28 | 75.14 | 78.55 |
> > | PromptSRC [ICCV'23] | 84.26 | 76.10 | 79.97 |
> > | MMA [CVPR'24]       | 83.20 | 76.80 | 79.87 |
> > | TCP [CVPR'24]       | 84.13 | 75.36 | 79.51 |
> > | DeKg [ICLR'25]      | 84.96 | 76.38 | 80.44 |
> > | MAIL [Ours]         | 85.19 | 77.39 | 81.10 |
> >
> > While our method achieves comparable accuracy to DeKg (ICLR’25) on the **base** classes, it yields a **significant** improvement on the **new** classes. The “new” setting refers to test classes that are not present during training, thereby placing higher demands on the model’s generalization ability—precisely where our MAIL excels.
> >
> > Therefore, we believe the improvements are both **statistically meaningful and practically valuable**.
>
>
>
> **R-2: Weakness-2 & 3**
>
> We thank the reviewer for the constructive feedback. We acknowledge that the y-axis not starting at zero may lead to visual misinterpretation. While our intention was to emphasize subtle performance differences, we agree that this could be misleading. In future versions, we will ensure to follow best practices for visualizations by starting the y-axis from zero and clearly indicating the axis range.
>
> Regarding Figure 1, our MAIL not only uses fewer learnable parameters than MaPLe and MMA but also achieves higher mAP. Although the training time is longer, the reduced parameter count results in fewer model weights to store, which is advantageous for real-world deployment. Importantly, we did not intentionally use the lower parameter count to obscure the longer training time—these two aspects are not equivalent, and suggesting so would be misleading. Interestingly, Reviewer-1 (Reviewer hW62) mentioned a related work, SVFT [3], an improvement over LoRA, which also employs fewer trainable parameters but requires longer training time.
>
>
>
> **R-3: Weakness-4 & Question-1**
>
> As discussed in the paper, training time and memory usage are the main limitations of our method. From our perspective, it is not always feasible to design an algorithm that simultaneously achieves lower training time, faster inference, and higher accuracy. We hope the reviewer understands this trade-off and the challenges it presents.
>
> Here we illustrate the reasons why our MAIL is still valuable though having more training time.
>
> > 1. In the context of vision-language models (VLMs), studying **modality coupling** is inherently valuable. Its importance becomes even more evident in few-shot scenarios, where it has already demonstrated strong effectiveness. We believe the reviewer has recognized this point.
> > 2. MAIL fills an important research gap—namely, the lack of **modality coupling** exploration into the third type of PEFT methods, i.e., partially fine-tuned approaches. This contribution is meaningful, as it can serve as a foundation and reference for future research. We do agree that training efficiency is important, and we also agree that how to reduce the training time can be more challenging, but we feel that our work represents an important starting point for further deeper analysis. This paper is not the end of the story, but rather a contribution to the broader line of research on modality coupling within the partially fine-tuned PEFT community.
> > 3. While MAIL may involve longer training time, the more important is, our MAIL has achieved remarkable performance, as responded in R-1.  It's a worth trade-off.
> > 4. What's more, compared to UCDR, FS-UCDR (which is also first proposed in this paper) already achieves a substantial reduction in training time. Therefore, we argue thats increase introduced by MAIL would not become a bottleneck.
>
>
>
> **R4:  Question-2**
>
> Thank you for the reviewer’s insightful comment. We would first like to clarify why MaPLe, despite having the largest number of parameters, exhibits the highest efficiency. Then, we explain why MAIL is slower than MMA.
>
> > 1. Regarding the efficiency of MaPLe [1]: The additional parameters in MaPLe come from extending the input sequence length rather than introducing new computational operations. These extended tokens are processed within the Transformer through matrix multiplications, which are highly parallelizable. As a result, the increase in computational workload is efficiently handled by modern hardware. In contrast, both MMA and our proposed MAIL introduce additional operations *within* each Transformer block. These operations are inherently sequential relative to the standard forward pass and cannot be easily parallelized. Therefore, both MMA and MAIL experience longer training times.
> > 2. Why MAIL is slower than MMA: MAIL performs its operations at four local positions within each Transformer block, whereas MMA applies its operation at a single global position. This design in MAIL leads to more frequent sequential computations, resulting in longer training time compared to MMA.
>
>
>
> Finally, we kindly ask the reviewer not to assess our method based solely on efficiency. As noted in the appendix (and also explicitly stated in the Limitations section of the main paper), we acknowledge the shortcomings of our approach, namely the training time and memory usage. However, given that the few-shot tuning paradigm for vision-language models is generally efficient and time-saving, we believe training time is unlikely to pose a significant bottleneck in practical applications.
>
>
>
> [1] Muhammad Uzair Khattak, Hanoona Rasheed, et al. Maple: Multi-modal prompt learning. CVPR, 2023.
>
> [2] Lingxiao Yang, Ru-Yuan Zhang, et al. Mma: Multi-modal adapter for  vision-language models. CVPR, 2024.
>
> [3] Lingam, Vijay Chandra, et al. SVFT: Parameter-efficient fine-tuning with singular vectors. NeurIPS, 2024.

---

### Official Review · Reviewer_hW62 · 2025-07-03

**Clarity:** 3
**Significance:** 3
**Originality:** 3
**Rating:** 4
**Confidence:** 2

**Summary:**

This paper proposes MAIL, a method for Few-Shot Unsupervised Cross-Domain Retrieval using coupled agent layers with scaling-and-shifting reparameterization to enhance CLIP's cross-modal alignment. MAIL maintains CLIP’s inference efficiency while improving adaptability, outperforming baselines on three benchmarks. However, its sequential processing incurs minor training and memory overhead.

**Questions:**

No

**Ethical Concerns:**

["NO or VERY MINOR ethics concerns only"]

**Final Justification:**

I am not an expert in this topic. The authors' response has addressed my concerns, and I will maintain my score.

**Limitations:**

Yes

**Quality:**

3

**Strengths And Weaknesses:**

Strengths

1. The coupled agent layers concept leveraging scaling-and-shifting reparameterization is an interesting approach. Its key strength is enhancing adaptation without compromising CLIP's original inference efficiency.
2. The propoesd method demonstrates consistent superiority over competitors across several benchmarks, provides evidence for MAIL's effectiveness in the Few-Shot Unsupervised Cross-Domain Retrieval setting.

Weaknesses

1. Relying on CLIP’s pre-trained features, MAIL may underperform in domains with unique visual-linguistic distributions (e.g., medical imaging or remote sensing), where pre-trained representations lack task-specific nuances. This limits adaptability in specialized fields.
2. I would like to see the performance comparision of MAIL with a similar PEFT methods named SVFT [1]. SVFT updates the weight as a sparse combination a diagonala matrix outer products of its singular vectors, training only the coefficients. The proposed MAIL learns a diagonal matrix with a scaling vector as its diagonal elements.

[1] Lingam, Vijay Chandra, et al. Svft: Parameter-efficient fine-tuning with singular vectors. Advances in Neural Information Processing Systems, 2024.

---

> ### Author Rebuttal · Authors · 2025-07-29
>
> **R-1: Weakness1**
>
> We appreciate the reviewer’s insightful comment. While it is true that CLIP’s pre-trained features can be suboptimal in highly specialized domains such as medical imaging or remote sensing, this limitation is not specific to MAIL, but rather a general challenge faced by all CLIP-based methods.
>
> Importantly, MAIL is designed as a **plug-and-play framework** that builds on top of CLIP and enhances its adaptability. Moreover, it can be easily integrated with other foundation models that are pre-trained on medical or remote sensing images, potentially leading to even better performance in such specialized domains.
>
> **R-2: Weakness2**
>
> Thank you for bringing the paper SVFT [1] to our attention. We agree that it is relevant to our work, and we will be happy to cite and include it in the discussion of our paper. Attempting to generalize our MAIL to the sparse matrix (instead of the diagonal matrix) is an interesting idea for future work.
>
> > To further acknowledge SVFT's contribution, we conducted experiments using two variants of SVFT, namely SVFT$^P$ and SVFT$^R_{d=12}$. Below, we report their FS-UCDR (2-shot) evaluation results (%) on DomainNet.
> >
> > | Method                      | Sketch |       |       |       | Quickdraw |       |  |       | Painting |       |       |       |
> > | --------------------------- | ------------- | ----- | ----- | ----- | ----- | ----- | ----- | ----- | ----- | ----- | ----- | ----- |
> > |                             | Unseen        |       | Mixed |       | Unseen |       | Mixed |       | Unseen |       | Mixed ||
> > |                             | map           | prec  | map   | prec  | map   | prec  | map   | prec  | map   | prec  | map   | prec  |
> > | ProS* [CVPR'24]             | 64.67         | 60.01 | 58.43 | 54.63 | 28.42 | 25.44 | 23.18 | 21.27 | 75.16 | 69.55 | 71.20 | 66.12 |
> > | CLIP [ICML'21]              | 42.20 | 35.28 | 36.62 | 29.79 | 7.44 | 5.61 | 6.00 | 3.17 | 61.68 | 55.07 | 56.53 | 50.14 |
> > | LORA [ICLR'22]              | 54.85 | 49.23 | 48.71 | 43.33 | 22.16 | 18.10 | 17.73 | 14.21 | 71.46 | 65.10 | 66.81 | 60.81 |
> > | SVFT$^P$[NeurIPS'24]        | 57.72 | 51.94 | 50.04 | 47.01 | 22.32 | 18.88 | 18.12 | 15.71 | 69.99 | 63.11 | 65.82 | 60.00 |
> > | SVFT$^R_{d=12}$[NeurIPS'24] | 60.74 | 53.41 | 53.82 | 51.90 | 24.98 | 20.39 | 19.40 | 17.63 | 73.95 | 67.37 | 68.44 | 63.33 |
> > | MAIL [Ours]                 | 65.76 | 61.57 | 59.05 | 55.25 | 29.41 | 26.95 | 22.83 | 21.26 | 76.05 | 70.85 | 71.12 | 66.44 |
> > | **Method**                  | **Infograph** |       |       |       | **Clipart** |       |       |       | **Average** |       |       |       |
> > |  | Unseen | | Mixed | | Unseen | | Mixed | | Unseen | | Mixed | |
> > |                             | map           | prec  | map   | prec  | map   | prec  | map   | prec  | map   | prec  | map   | prec  |
> > | ProS* [CVPR'24]             | 57.98         | 54.42 | 52.19 | 49.56 | 76.48 | 71.86 | 72.28 | 68.15 | 60.52 | 56.26 | 55.46 | 51.95 |
> > | CLIP [ICML'21]              | 50.08 | 44.74 | 43.75 | 38.91 | 60.37 | 51.30 | 56.08 | 46.91 | 44.35 | 38.40 | 39.80 | 33.78 |
> > | LORA [ICLR'22]              | 58.01 | 53.84 | 51.86 | 48.20 | 70.52 | 64.11 | 66.00 | 59.63 | 55.40 | 50.08 | 50.22 | 45.24 |
> > | SVFT$^P$[NeurIPS'24]        | 58.47 | 52.55 | 52.13 | 48.65 | 71.77 | 65.94 | 67.33 | 60.57 | 56.05       | 50.48 | 50.69 | 46.39 |
> > | SVFT$^R_{d=12}$[NeurIPS'24] | 57.96 | 52.04 | 51.30 | 47.26 | 72.12 | 66.56 | 69.02 | 62.25 | 57.95       | 51.95 | 52.40 | 48.74 |
> > | MAIL [Ours]                 | 60.11 | 57.40 | 53.34 | 50.95 | 78.94 | 74.80 | 73.91 | 70.14 | 62.05 | 58.31 | 56.05 | 52.80 |
> >
> >As shown in the table, SVFT performs worse than our proposed MAIL. Interestingly, while SVFT$^R_{d=12}$ achieves performance comparable to LoRA in the original SVFT paper, it exhibits a more significant improvement over LoRA under the FS-UCDR setting.
>
> So far, LoRA-based methods have seen limited application in the context of CLIP. As shown in our Table 4 （from the main paper）, the current state-of-the-art methods for few-shot classification are all PEFT approaches that are not based on LoRA. This may be because LoRA has been primarily explored in the domain of large language models, while its adaptation to CLIP remains under-explored. As a result, directly applying LoRA to CLIP without further optimization may lead to suboptimal performance.
>
> [1] Lingam, Vijay Chandra, et al. SVFT: Parameter-efficient fine-tuning with singular vectors. Advances in Neural Information Processing Systems, 2024.

---

### Note · Authors · 2025-08-13

We sincerely thank the AC and all reviewers for their valuable time and constructive feedback.

**Key recognitions from the reviews:**

1. The idea of MAIL is interesting. (**hW62**, **ZXLZ**)
2. MAIL incurs no additional inference overhead. (**hW62**, **ZXLZ**)
3. MAIL achieves competitive performance on both FS-UCDR and few-shot classification benchmarks. (**All reviewers**)
4. The newly introduced FS-UCDR scenario is practical. (**8mcV**)

---

We summarize our rebuttal as follows:

### For **Reviewer hW62**

We have tried our best to address the following concerns:

1. **Underperformance in domains with unique visual-linguistic distributions**

   * MAIL is a **plug-and-play** framework built on CLIP, enhancing adaptability and easily extendable to other foundation models (e.g., medical or remote sensing), where it may perform even better.
2. **Comparison with SVFT**

   * MAIL outperforms SVFT by a large margin.

### For **Reviewer ZXLZ**

We have tried our best to address the following concerns:

1. **Performance vs. MMA with similar tunable parameters**

   * The performance gap is significant, as shown in our results.
2. **Intrinsic interest of MAIL**

   * MAIL fills an important gap by introducing **modality coupling** into partially fine-tuned PEFT methods—a novel contribution that can serve as a foundation for future research. While training efficiency is important, we view this work as a meaningful starting point rather than a final solution.
---
It is unfortunate that no further feedback has been received from **hW62** and **ZXLZ** since our last response. We hope the AC and the reviewer can discuss further, and look forward to a positive outcome.

---

### For **Reviewer 8mcV**

We appreciate the constructive exchange and the score increase. We will further improve the paper by:

* Adding a clearer motivation for MAIL, explaining parameter alignment from a regularization perspective.
* Defining “original layer,” “target layer,” and “agent layer” in the Introduction.

---

We again thank the AC and all reviewers for their time and effort, and we look forward to the final decision.

---

### Decision · Program_Chairs · 2025-09-17

**Decision:**

Accept (poster)

**Comment:**

This submission received diverse borderline scores: 2xBA and 1xBR (with some upwards dynamic after the rebuttal). Overall, the reviewers appreciated introduction of MAIL, with fairly strong and consistent performance gains across 11 few-shot benchmarks and no added inference-time complexity. The raised concerns were mostly around the method and the implementation being fairly complicated, and insufficient motivation for the method overall and for specific components (like different types of layers). The rebuttal from the authors was thorough, and addressed many of those concerns effectively. The remaining points of the reviewer who gave the BR rating were about presentation, also with certain residual skepticism around the magnitude of the performance gains compared to the closest baseline.

After reviewing the paper and the discussion, AC agrees with the authors that the performance gains are convincing and meaningful across a broad set of benchmarks, and the method will likely only benefit from stronger pretraining of the base features in specialized domains. The final recommendation is therefore to accept. However, the authors are strongly encouraged to make sure that all remaining concerns around motivation and presentation are fully addressed in the camera ready version.